# EVA: Bridging Performance and Human-Alignment in Hard-Attention Vision Models

## Abstract

Humans recognize images by actively sampling them through saccades and fixations. Hard attention models mimic this process but are typically judged only on accuracy. We introduce EVA, a brain-inspired hard-attention vision model designed to deliver strong classification performance while simultaneously producing human-aligned gaze patterns and interpretable internal dynamics. EVA operates with a small number of sequential glimpses, combining a human-inspired foveal-peripheral glimpse module, neuromodulator-based variance control, and a gating mechanism. On the image classification benchmark CIFAR-10, for which human gaze data is available, we show that EVA achieves a compelling trade-off between accuracy and scanpath similarity, comparable to efficient CNNs and other hard attention baselines. Crucially, we demonstrate that EVA's learned fixation policy aligns with human scanpaths across multiple metrics (NSS, AUC). Further, its internal recurrent states yield class-specific trajectories in PCA space, revealing structured, interpretable processing dynamics. Ablation studies show that while the CNN backbone drives performance, the gating and neuromodulator modules uniquely enable alignment and interpretability. These results suggest that combining brain-inspired structural modules can yield vision models that are not only efficient and accurate but also transparent and human-aligned, a step toward jointly advancing performance and interpretability.

## 1 Introduction

The deployment of artificial intelligence towards application requires not only strong performance but also *interpretability* and *reliability*. If humans cannot understand the reasoning of AI systems, trust and safe integration into social contexts remain limited. Post-hoc explanation techniques, such as saliency maps or Grad-CAM applied to Convolutional Neural Networks (CNNs) or Vision Transformers (ViTs), have shown to be fragile and often misleading (LeCun et al., 2015; Wu et al., 2022; Rudin, 2019; Selvaraju et al., 2017). Thus, the challenge is not only to explain black-box models after the fact, but to design models whose internal mechanisms are inherently interpretable.

Human vision provides a natural blueprint. Due to biological constraints, perception is inherently selective: we cannot process the entire visual field at high resolution simultaneously. Instead, we rely on a sequence of saccadic eye movements, shifting the fovea to sample informative regions of a scene (Desimone & Duncan, 1995; Yarbus, 1967; Rayner, 1998). These gaze samples form the foundation of human visual attention enabling efficient perception. Inspired by this principle, hard attention models (Mnih et al., 2014; Williams, 1992) attempt to replicate the process by selecting glimpses and learning fixation policies through reinforcement learning. To achieve interpretability, one can naturally consider measuring the alignment of scanpaths, sequences of sampled visual attention locations, between the hard attention models and humans.

We view gaze as a form of sampling behavior, analogous to language. Wittgenstein's language games highlight how meaning arises not from isolated symbols, but from patterns of use within a shared context (Wittgenstein, 1953). Similarly, gaze trajectories are samples shaped by individual perception yet constrained by the shared task and environment (Yarbus, 1967). Therefore, we choose image classification as our primary task, though the approach can be extended to broader domains such as object detection task, or robotics (Kim et al., 2021; 2020). Drawing on this point, we propose the hypothesis that when an AI system's attention mechanism exhibits structural characteristics that

are analogous to human visual fixation behaviour such as gating and prediction error modulation, it will not only improve task performance under resource constraints, but also yield internal decision-processes that are more readily interpretable by human observers. In particular, even though individual human scanpaths contain noise and inter-observer variability, the aggregate structural correspondence between model fixations and human fixations may serve as a proxy for alignment of processing strategy, similar to Borji & Itti (2014); while specifically, in image recognition task, human can see a decisive glimpse evidence in a moment when the classifier is changing from one prediction to another, thereby increasing trust and transparency of the model. (Fig. 1).

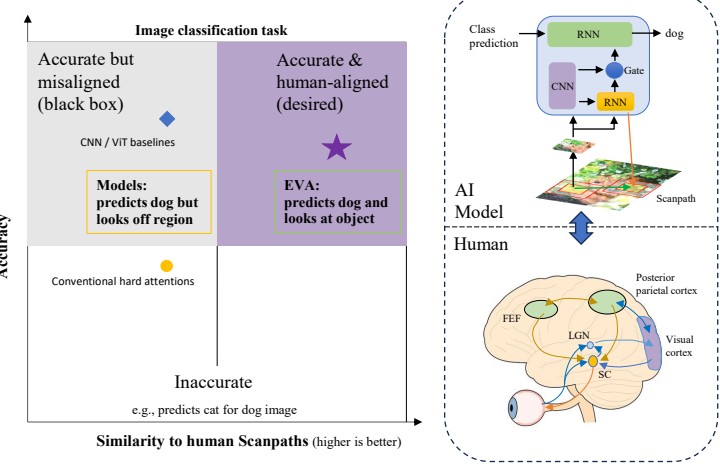

Figure 1: A path toward interpretable vision: from brain-inspired structure to behavior alignment. **Left**: trade-off between accuracy and similarity to human scanpaths on an image classification task. Conventional CNN/ViT baselines are accurate but behaviorally misaligned, while conventional hard-attention models are less accurate and still misaligned. EVA aims for the upper-right region: accurate and human-aligned. Axes are schematic and not drawn to scale. **Right**: schematic correspondence between EVA and oculomotor circuitry. Matching colors indicate qualitative correspondence between model modules and brain areas.

Humans balance exploration and exploitation by fixating on informative regions while also making long-range saccades when uncertainty or prediction error requires new evidence (Renninger et al., 2007; Feldman & Friston, 2010; Kujala & Lappi, 2021; Dayan & Yu, 2002). Neuroscience indicates that this process is supported by a dual architecture: the collicular pathway enables rapid eye-movement control via the superior colliculus (SC), while cortical pathways integrate bottom-up retinal input with top-down modulation from the pulvinar and visual cortex (Kandel et al., 2000; Fischer & Whitney, 2012). Motivated by this biological structure, we introduce three brain-inspired components into a multilayer hard attention model (Pan et al., 2025): (1) a CNN module that processes foveated glimpses, (2) a neuromodulatory mechanism that adjusts saccadic variability under uncertainty, and (3) a pulvinar-inspired gate that regulates the flow of visual information.

In this work, we introduce **EVA**, a lightweight brain-inspired hard attention model trained solely on classification labels that produces **E**mergent human-like **V**isual **A**ttention. EVA achieves compelling performance among hard attention baselines and competitive accuracy relative to soft-attention models, while generating scanpaths that closely align with human gaze data. To our knowledge, this is the first systematic evaluation of hard attention glimpse trajectories against human eye movements, offering both efficiency and a novel path toward interpretable AI.

By synchronizing the visual attention of models and humans, we enable a new dimension of interpretability: decisions can be understood not through abstract feature maps, but by observing that the model "looks where humans look." Prior work has correlated model explanations with human gaze in driving tasks (Hwu et al., 2021), and in robotics, gaze has long been used as a communicative signal (Admoni & Scassellati, 2017; Lara Naendrup-Poell, 2025). Such synchronization can foster human–AI communication and collaboration, providing common ground akin to shared gaze in social interaction.

## 2 Related Work

### 2.1 Brain-inspired Vision Models

Deep CNNs trained on object recognition have become influential models of the primate ventral visual pathway (Lindsay, 2021; Xu et al., 2015). A series of studies showed a striking layer-wise correspondence: features from early CNN layers resemble V1/V2 representations (edge and texture filters), while deeper layers provide the best predictions of neural activity in mid-level and inferotemporal (IT) cortex. These findings suggest that feedforward CNN architectures capture core hierarchical vision computations, and they have spurred benchmarks like Brain-Score to quantify "brain-likeness" of networks. Notably, some architectures optimized for both task accuracy and neural alignment (e.g. CORnet-S, a recurrent CNN) score highly on such benchmarks Schrimpf et al. (2020). However, conventional CNN-based models typically lack the brain's top-down attentional modulation and gating mechanisms. In biological vision, extensive feedback and structures like the pulvinar thalamus dynamically route information between areas during perception aspects not captured by feedforward CNNs alone (Fischer & Whitney, 2012; Purushothaman et al., 2012; Beck & Kastner, 2009; Cao et al., 2015). This gap motivates hybrid models that integrate CNN front-ends with neuro-inspired attention controllers.

### 2.2 Hard Attention

Early hard attention models in vision make classification decisions using a sequence of glimpses, each focusing on a subset of image pixels. In RAM, a neural controller (RNN) chooses successive regions to attend, inspired by the human foveation mechanism where only a small high-resolution region is seen at a time and eye movements sequentially sample the scene (Larochelle & Hinton, 2010; Butko, 2009; Mnih et al., 2014). However, training such hard-attention policies is challenging because the non-differentiable glimpse selection must be learned via reinforcement signals like REINFORCE (Williams, 1992), leading to high variance and convergence difficulties. Consequently, early hard-attention models were demonstrated mostly on simpler datasets (MNIST, SVHN) and struggled to scale to complex imagery. Extensions like the Deep Recurrent Attention Model (DRAM) enabled multiple glimpse scales or multi-object reasoning, and showed improved performance on tasks like multi-digit classification with a global context over extracted by over a full-image CNN (Ba et al., 2014). More recent efforts like Saccader introduced better training strategies (e.g. patch-wise pretraining) to narrow the performance gap on ImageNet while selecting just a few informative regions (Elsayed et al., 2019). Recent work on active vision include a predictive coding and uncertainty minimization method to decide where to look, yielding efficient exploration and unsupervised scene understanding, and a multi-layer RAM (MRAM) model that separates visual understanding and action using 2 RNNs(Sharafeldin et al., 2024; Pan et al., 2025). Still, standard end-to-end hard-attention models that rely only on the limited glimpse lack certain biological considerations for modeling and evaluating active visual observer like human and scale up to real-world environment.

### 2.3 Human Gaze Alignment with Artificial Models

A central question in human–AI alignment for vision is whether the internal attention mechanisms of artificial systems correspond to human gaze patterns when viewing images or performing visual tasks. Classical saliency models and modern deep saliency networks can predict where people tend to fixate in free-viewing of natural scenes, but they typically produce static fixation maps rather than full sequential scanpaths (Itti et al., 1998; Qiao et al., 2018). Early studies have nonetheless shown that human gaze can serve as a useful supervisory signal, improving performance in visual recognition and robotics when used as an additional training cue (Kim et al., 2021; 2020; Li et al., 2025). More recently, large-scale gaze datasets on standard computer-vision benchmarks such as COCO-Search18 and Gaze-CIFAR10 have enabled systematic evaluation of scanpath prediction models (Chen et al., 2021; Yang et al., 2020; Yang & Samaras, 2022; Li et al., 2025). Contemporary approaches explicitly learn to predict human scanpaths from images using powerful deep architectures: transformer-based soft-attention encoders, diffusion models, and other sequence generators trained directly on eye-movement data (Linardos et al., 2021; Yang et al., 2024; Mondal et al., 2023; Cartella et al., 2025; Chen et al., 2024). These top-down models treat human gaze as a target output to be fitted. In contrast, our work asks a complementary bottom-up question: *can a task-driven vision system that is never trained on human gaze nevertheless develop human-like scanpaths as an emergent behavior?* We

study a hard-attention model whose foveated policy is guided by brain-inspired mechanisms, and we evaluate whether its closed-loop glimpse trajectories naturally reproduce the spatial–temporal characteristics of human saccades during image recognition task.

## 3 METHOD

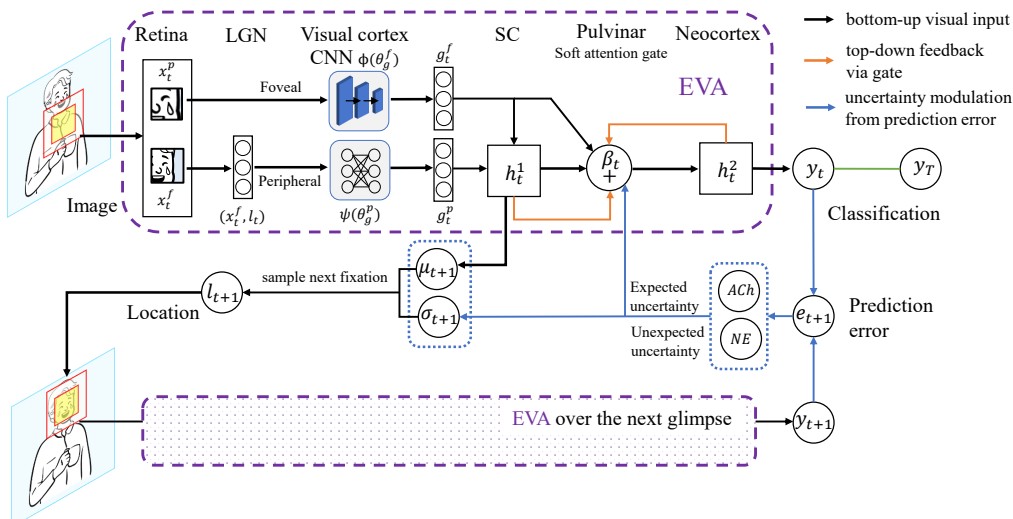

Figure 2: EVA model architecture. **Top**: one glimpse step at time t. Foveal and peripheral crops from the retina, with foveal-only processed by a visual-cortex CNN, are combined in a superior-colliculus (SC)–like recurrent unit and a pulvinar-inspired soft attention gate, and then integrated in a neocortex-like recurrent classifier. **Bottom**: the prediction error modulates the fixation variance and gate.

Hard attention models emulate human visual perception by sequentially selecting informative image regions rather than processing the entire image at once. However, prior works such as RAM (Mnih et al., 2014) and MRAM (Pan et al., 2025) lack effective mechanisms for high-level visual processing. In addition, neuroscience shows that humans regulate attention not only bottom-up, driven by stimulus salience, but also top-down, guided by cortical feedback that suppresses or enhances sensory signals (Beck & Kastner, 2009; Desimone, 1998; Dayan & Yu, 2002; Miller et al., 2018). Directly adding a powerful CNN to RAM often destabilizes training, since the network can rely on CNN features without learning a meaningful saccade policy. To address this imbalance, we incorporate three brain-inspired components: (i) a CNN module resembling visual cortex for foveal vision, (ii) a neuromodulator that adapts saccadic variability based on prediction error, and (iii) a pulvinar-inspired gate that selectively routes information between recurrent layers. Together, these modules synchronize bottom-up perception with top-down modulation.

### 3.1 OVERVIEW OF RAM AND MRAM

RAM and its extension MRAM consist of four main modules: a *glimpse module*, a *core recurrent module*, a *location module*, and an *action module*. The glimpse module extracts features from a local image patch, mimicking foveal vision. The recurrent module integrates these glimpses over time. The location module predicts the next fixation, while the action module makes the classification decision. EVA (MRAM backbone) separates these roles across a two-layer recurrent architecture. The lower RNN predicts fixation locations, while the upper RNN integrates evidence for classification. At each timestep $t$, given a foveal crop $x_t^f$ and peripheral context $x_t^p$, the model computes:

$$g_t = f_g\Big([x_t^f \| x_t^p], l_t; \theta_g\Big), \quad h_t^1 = f_h^1(h_{t-1}^1, g_t; \theta_h^1), \quad h_t^2 = f_h^2(h_{t-1}^2, h_t^1; \theta_h^2), \tag{1}$$

$$l_{t+1} \sim p(\cdot \mid f_l(h_t^1; \theta_l)), \qquad a_t = f_a(h_t^2; \theta_a). \tag{2}$$

Here $g_t$ is the glimpse representation, $h_t^1$ and $h_t^2$ are the lower and upper RNN states, $l_{t+1}$ is the fixation location sampled from a stochastic distribution, and $a_t$ is the classification output.

## 3.2 Visual Cortex–Inspired CNN Module

In EVA, to mimic cortical processing of foveal input, we introduce a CNN $\phi(\cdot)$ that processes the central crop $x_t^f$, and the same fully connected layer in RAM, $\psi(\cdot)$ process the raw peripheral view $x_t^p$ and the current fixation $l_t$. Their outputs are concatenated to form the glimpse feature $\mathbf{s}_t$:

$$g_t^f = \phi\Big(x_t^f; \theta_g^f\Big), \quad g_t^p = \psi\Big(x_t^p, l_t; \theta_g^p\Big), \quad \mathbf{s}_t = \big[g_t^f \| g_t^p\big]. \tag{3}$$

The resulting features $\mathbf{s}_t$ are passed to the lower RNN:

$$h_t^1 = f_1(h_{t-1}^1, \mathbf{s}_t). \tag{4}$$

## 3.3 Neuromodulator with Prediction Error

The location module predicts the next fixation as:

$$l_{t+1} \sim \mathcal{N}(\mu(h_t^1), \sigma_t^2). \tag{5}$$

In RAM, variance $\sigma_t$ is fixed. Here, we adapt $\sigma_t$ dynamically using a neuromodulator inspired by acetylcholine (ACh) and norepinephrine (NE) systems (Dayan & Yu, 2002). We compute long-term and short-term exponential moving averages (EMAs) of prediction error:

$$\bar{e}_t^{(k)} = \tau_k \bar{e}_{t-1}^{(k)} + (1 - \tau_k)e_t, \quad k \in \{\text{long}, \text{short}\}, \tag{6}$$

where $e_t$ is instantaneous error, and $\tau_{\text{long}} > \tau_{\text{short}}$. The uncertainty signal is

$$u_t = \big|\bar{e}_t^{\text{s}} - \bar{e}_t^{\text{l}}\big|. \tag{7}$$

Finally, $\sigma_t$ is bounded using:

$$\sigma_t = \sigma_{\min} + (\sigma_{\max} - \sigma_{\min})\tanh(\alpha u_t). \tag{8}$$

Thus, high uncertainty increases fixation variance, encouraging exploration, while low uncertainty stabilizes fixation.

## 3.4 Pulvinar Gate

The pulvinar modulates information flow between SC and cortex. Analogously, we introduce a gate $\beta_t$ between the lower RNN $h_t^1$ and upper RNN $h_t^2$, implemented as a small QKV-style attention module, governed by their activations and the global uncertainty $\sigma_t$:

$$\beta_t = \text{clamp}\Big((f_{td}(h_{t-1}^2) - \sigma_t) \odot (f_{bu}(h_{t-1}^1) + \sigma_t), \, 0, 1\Big), \tag{9}$$

$$\bar{\beta}_t = \gamma \bar{\beta}_{t-1} + (1 - \gamma)\beta_t. \tag{10}$$

Both $f_{td}$ and $f_{bu}$ are implemented as single-layer MLPs with ReLU activations. This gate balances bottom-up visual evidence and top-down feedback. We further project $Q, K, V$ features and compute gated updates (Eqs. 11–15). Intuitively, when uncertainty is high, the gate admits more cortical input, aligning exploration with integration. The attention-style gate naturally implements multiplicative, normalized gain control over lower-level features conditioned on higher-level expectations, which matches the proposed function of the pulvinar as a dynamic relay of cortico–cortical communication.

$$Q_t = W_Q \, h_{t-1}^2, \tag{11}$$

$$K_t = W_K\big[h_t^1 \, \| \, \phi(x_t^{\text{f}})\big], \quad V_t = W_V \, K_t, \tag{12}$$

$$\alpha_t = \text{sigmoid}\Big(\tfrac{1}{\sqrt{d}} \, Q_t K_t^\top\Big), \tag{13}$$

$$\mathbf{z}_t = (1 - \bar{\beta}_t) \odot \big(\alpha_t V_t + \varepsilon V_t\big), \qquad \rho_t = \bar{\beta}_t \odot h_{t-1}^2, \tag{14}$$

$$h_t^{\text{h}} = \big[\mathbf{z}_t \, \| \, \rho_t\big]. \tag{15}$$

## 3.5 TRAINING

We follow a policy gradient approach for learning the location policy akin to the same REINFORCE algorithm applied in RAM and other hard attention models Mnih et al. (2014); Williams (1992).

The policy gradient loss via REINFORCE maximizes the expected reward. The reward $R$ is given 1 if the classification is correct, 0 otherwise. We subtract a baseline $b_t$ same in Pan et al. (2025) to reduce variance:

$$\mathcal{L}_{\text{REINFORCE}} = -\sum_{t=1}^{T} (R - b_t) \log \pi(\ell_t \mid h_t^1; \theta), \tag{16}$$

where $\pi(\ell_t \mid h_t^1; \theta)$ is the policy distribution based on hidden state of the lower recurrent layer $H_t^1$. In addition, we include a standard supervised classification loss (cross-entropy) on the final classifier output:

$$\mathcal{L}_{\text{CE}} = -\sum_{t=1}^{T} y_T \, \log y_t, \tag{17}$$

where $y_t$ is the predicted probability of class $c$ and $y_T$ is the one-hot indicator of the true label.

The full training loss becomes:

$$\mathcal{L} = \mathcal{L}_{\text{CE}} + \mathcal{L}_{\text{REINFORCE}} + \lambda_{\text{cost}} \, \overline{\beta}_b + \lambda_1 \, \|\overline{\beta}_b\|_1 + \lambda_H \, \mathcal{H}_\beta, \tag{18}$$

where $\overline{\beta}_b$ is the mean bottom-up gate value, and the entropy term

$$\mathcal{H}_\beta = -\overline{\beta}_t \log(\overline{\beta}_t + \varepsilon) - \overline{\beta}_b \log(\overline{\beta}_b + \varepsilon) \tag{19}$$

encourages non-trivial gating behavior. Hard attention remains challenging to train, to stabilize the neuromodulator, we used prediction error with ground-truth labels during training, then switched to prediction error between glimpses (self-error) at test time. We observed $\sim 5\%$ accuracy drop when trained purely with self-prediction error (train self-error), suggesting that using label-based error during training stabilizes learning.

## 3.6 COMPOSITE SCANPATH SIMILARITY METRIC

While individual scanpath metrics provide complementary perspectives, no single metric fully captures alignment between human and model gaze. In our evaluation, we therefore combined four standard measures into a composite *scanpath similarity* (SS) score: Dynamic Time Warping (DTW), ScanMatch (SM), Normalized Scanpath Saliency (NSS), and Area Under the Curve (AUC) (DTW, 2007; Peters et al., 2005; Cristino et al., 2010; Zanca et al., 2018).

DTW quantifies sequential similarity between fixation trajectories. ScanMatch encodes scanpaths as symbolic sequences and measures overlap. NSS evaluates how well model-predicted fixations fall on human-derived saliency maps, and AUC measures fixation prediction in a probabilistic sense. For NSS and AUC, we constructed continuous saliency maps by convolving fixation points (12 glimpses per trial) with a Gaussian kernel ($\sigma = 7$).

We then aggregated these metrics into a single SS score using weighted averaging:

$$SS = W_{DTW} \, \mathcal{D} + W_{SM} \, \mathcal{S} + W_{NSS} \, \mathcal{N} + W_{AUC} \, \mathcal{A}, \tag{20}$$

where $\mathcal{D}, \mathcal{S}, \mathcal{N}, \mathcal{A}$ denote *normalized* DTW, ScanMatch, NSS, and AUC scores respectively. Each metric is first linearly rescaled to $[0, 1]$ across all compared models so that larger values indicate better alignment after normalization. The normalization procedure is detailed in Appendix D.1. We use equal weights $W_{DTW} = W_{NSS} = W_{AUC} = W_{SM} = 0.25$ to avoid over-emphasizing any single metric. The choice of weights is heuristic, and therefore we also report all four metrics individually alongside the composite score in Table 1.

## 4 EXPERIMENTS

### 4.1 PERFORMANCE ON IMAGE CLASSIFICATION BENCHMARKS

We first evaluate **EVA** on CIFAR-10 and ImageNet-10. These datasets allow us to jointly assess classification accuracy, parameter efficiency, and gaze alignment under consistent training settings.

Table 1: CIFAR-10: accuracy and scanpath similarity results. **Bold** marks the best performance at the metric. Light blue marks 2 variants of our proposed model, where EVA-Mobile is implemented with pretrained MobileNetV3.

| Model | Params (M) | Time (ms/im) | FLOPs (B)↓ | Acc. (%)↑ | DTW ↓ | SM ↑ | NSS ↑ | AUC ↑ | SS ↑ |
|---|---|---|---|---|---|---|---|---|---|
| CNN (ResNet18) | 11.18 | 0.89 ± 0.02 | 0.34 | 78.00 | - | - | - | - | 0 |
| CNN (MobileNetV3) | 4.21 | 0.94 ± 0.08 | 0.03 | 78.52 | - | - | - | - | 0 |
| TinyViT | 4.37 | 3.03 ± 0.22 | 2.66 | 68.21 | - | - | - | - | 0 |
| DeepGaze IIE | 0.2 | 0.82 ± 0.03 | - | - | 705.48 | 0.311 | 0.251 | 0.601 | 0.28 |
| Gazeformer | 0.27 | 0.04 ± 0.01 | - | - | **669.22** | **0.346** | 0.757 | 0.689 | **0.37** |
| Saccader | 12.49 | 3.29 ± 0.31 | 4.44 | 77.80 | 928.44 | 0.276 | 0.277 | 0.665 | 0.28 |
| RAM, 1scale | 0.65 | 2.21 ± 0.08 | 0.01 | 62.27 | 1176.54 | 0.241 | 0.228 | 0.654 | 0.24 |
| RAM, 2scale | 0.68 | 3.22 ± 0.20 | 0.02 | 61.55 | 1173.89 | 0.258 | 0.377 | 0.684 | 0.26 |
| DRAM, 1scale | 2.24 | 1.94 ± 0.10 | 1.74 | 64.81 | 1069.77 | 0.248 | 0.224 | 0.656 | 0.27 |
| DRAM, 2scale | 2.25 | 3.91 ± 0.35 | 1.75 | 62.17 | 837.14 | 0.302 | 0.665 | 0.679 | 0.31 |
| MRAM, 1scale | 1.18 | 2.34 ± 0.13 | 0.01 | 64.18 | 930.38 | 0.274 | 0.318 | 0.667 | 0.29 |
| MRAM, 2scale | 1.21 | 3.53 ± 0.28 | 0.02 | 58.24 | 945.31 | 0.263 | 0.315 | 0.674 | 0.28 |
| EVA (w/o CNN) | 1.93 | 3.12 ± 0.45 | 0.07 | 62.41 | 800.30 | 0.327 | 0.511 | **0.703** | 0.34 |
| EVA (CNN only) | 2.22 | 3.91 ± 0.09 | 1.32 | 69.99 | 1019.85 | 0.253 | 0.264 | 0.663 | 0.27 |
| EVA (w/o gate) | 2.48 | 2.96 ± 0.14 | 1.32 | 78.96 | 863.70 | 0.308 | 0.391 | 0.686 | 0.32 |
| EVA (gate only) | 1.67 | 2.92 ± 0.06 | 0.07 | 55.61 | 797.79 | 0.318 | 0.702 | 0.681 | 0.34 |
| EVA (w/o error) | 2.97 | 3.32 ± 0.11 | 1.37 | 75.14 | 894.17 | 0.300 | 0.386 | 0.691 | 0.31 |
| EVA (error only) | 1.47 | 3.24 ± 0.42 | 0.02 | 63.36 | 824.51 | 0.321 | 0.483 | 0.702 | 0.34 |
| EVA (train self-error) | 2.97 | 3.24 ± 0.35 | 1.37 | 73.78 | 792.89 | 0.330 | 0.608 | 0.700 | 0.35 |
| EVA-Mobile | 4.79 | 5.59 ± 0.49 | 0.45 | 76.14 | 825.11 | 0.322 | **0.786** | 0.701 | 0.35 |
| EVA | 2.97 | 3.06 ± 0.04 | 1.37 | **79.77** | 856.29 | 0.316 | 0.481 | 0.692 | 0.33 |

Baselines include convolutional models (ResNet18, MobileNetV3), a transformer (ViT-tiny), and hard-attention models (RAM, DRAM, MRAM, Saccader). All models share the same glimpse size, number of steps, and optimizer hyperparameters to ensure a fair comparison. Details of experiments are described in Appendix D

As summarized in Table 1, EVA attains the highest accuracy among hard-attention models on CIFAR-10, while using fewer FLOPs than ResNet18 and TinyViT. EVA-Mobile trades a small drop in accuracy for the best composite scanpath similarity (SS), illustrating the accuracy–alignment trade-off. Developed human attention prediction models such as Gazeformer trained directly on the Gaze-CIFAR-10 data, tends to produce temporal-aligned scanpaths where indicating by trajectory-level metrics like DTW and SM with strong salience metrics (NSS, AUC). However, EVA-Mobile model specifically, reached the best performance in saliency-based metrics, suggesting the model shared more similarity between human saliency than a attention prediction along. This underscores the limitation of saliency-only models when evaluated on dynamic scanpaths in task-related contents. This establishes a new reference point showing that reinforcement-based attention agents can approximate human gaze strategies in image classification.

Table 2: Accuracy (%) of hard-attention models under different gaze policies on CIFAR-10. The *Predicted* column reports the learned policy; other columns fix or perturb gaze trajectories. Numbers in parentheses indicate accuracy drop relative to Predicted.

| Model | Gaze Policy | | | | |
|---|---|---|---|---|---|
| | Predicted | Center-fixed | Corner-fixed ↓ | Random | Shuffled ↓ |
| RAM | 61.6 | 45.1 (-16.5) | 12.8 (-48.8) | 43.5 (-18.1) | 61.3 (-0.3) |
| DRAM | 62.2 | 41.6 (-20.6) | 11.6 (-50.6) | 38.4 (-23.8) | 61.9 (-0.3) |
| MRAM | 58.3 | 44.1 (-14.2) | 13.6 (-44.7) | 48.7 (-9.6) | 56.9 (-1.4) |
| **EVA** | 79.8 | 57.6 (-22.2) | 19.6 (-60.2) | 70.5 (-9.3) | 71.3 (-8.5) |

**Ablation studies.** Table 1 also reports controlled ablations to isolate the contributions of EVA's three biologically inspired modules. Removing the CNN (EVA w/o CNN) severely harms accuracy, confirming the importance of strong foveal features. The neuromodulator (EVA w/o error) modestly affects accuracy but reduces robustness and SS. Omitting the pulvinar gate (EVA w/o gate) slightly lowers accuracy and substantially weakens SS; using only the gate or only the error signal yields high SS but poor accuracy, indicating that these modules shape gaze behaviour but cannot replace the CNN. Training the neuromodulator with errors between EVA's predictions (train self-error) rather than true label will slightly reduce the performance. Finally, EVA-Mobile replaces the CNN with a pretrained MobileNetV3 backbone, yielding the highest SS at a small cost in accuracy. Overall, the three modules act synergistically: the CNN supports recognition, while neuromodulation and gating adapt exploration under uncertainty and balance bottom-up and top-down information flow.

**Stability under gaze perturbations.** To probe the interpretability of EVA's learned gaze policy, we fixed glimpse trajectories using artificial policies (center-only, corner-only, random, shuffled) as in Table 2. EVA suffers the largest accuracy drop when constrained to corner-fixed or shuffled policies, indicating that it relies on a task-aligned temporal-based sequence of saccades rather than localized bias. RAM and DRAM degrade sharply under random or corner-fixed policies but are almost unaffected by shuffled human scanpaths, suggesting that their policies are less tightly coupled to specific fixation orders. EVA's sensitivity to perturbations is therefore evidence that its learned scanpaths are functionally meaningful for recognition and interpretability (Appendix D.2).

## 4.2 EMERGENT HUMAN-LIKE GAZE PATTERNS

To probe whether model's scanpaths resemble human gaze as the SS metrics in Table 1, we visualized the model predicted scanpath to human. Draw on the CIFAR-10 image, this visualization can facilitate understanding on how a good scanpaths should be in image classification. Figure 3 visualizes top 5 models on the SS scores including Gazeformer, EVA, EVA-Mobile, Saccader, and DRAM predicted fixations alongside human gaze trajectories. For hard-attention models, these fixations are the key internal variables and are used solely to support image classification, while for Gazeformer, it's an optimization target. EVA is not a dedicated gaze-prediction model, but it is the hard-attention model whose scanpaths most closely align with human trajectories. EVA consistently focuses on semantically informative regions such as faces, wheels, and object centers. While discrepancies remain, partly due to upsampling artifacts in Gaze-CIFAR-10, EVA's trajectories typically overlap human fixations more closely than other hard-attention baselines. This offers a concrete form of interpretability: one can understand EVA's decisions not only through accuracy prediction, but with humans-like visual processing, suggesting shared attentional grounding for human-AI collaboration.

Figure 4 shows the joint comparison of classification accuracy and composite SS across hard-attention baselines. EVA achieves the best overall trade-off: it attains the highest accuracy while maintaining strong scanpath similarity, whereas EVA-Mobile slightly sacrifices accuracy for even higher SS. This highlights how biologically inspired mechanisms such as neuromodulation and pulvinar-like gating can simultaneously support task performance and emergent human-like gaze behaviour.

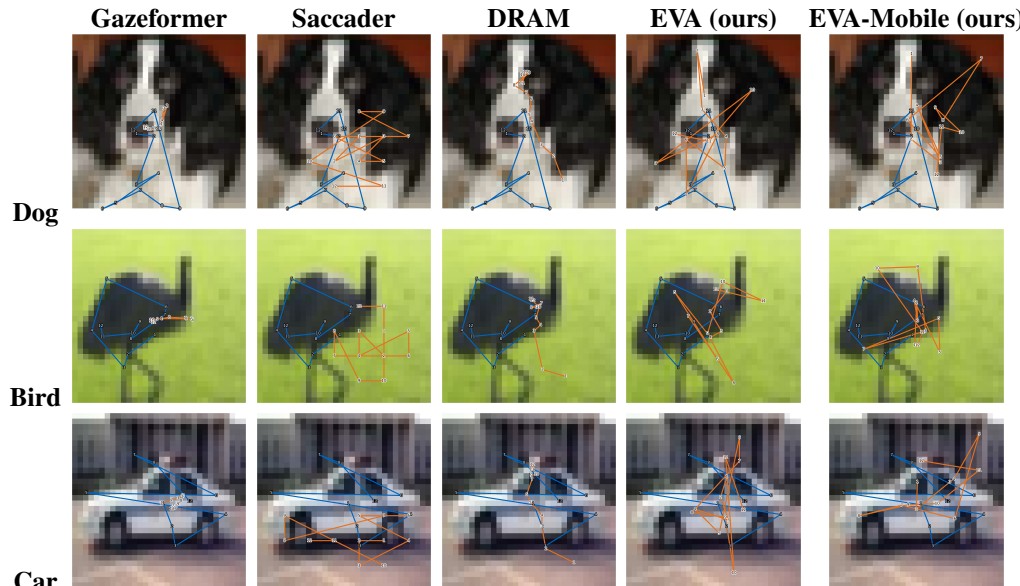

Figure 3: Qualitative comparison of human and model scanpaths on CIFAR-10. Columns: different models; rows: different example images. **Orange**: model scanpath; **Blue**: human scanpath.

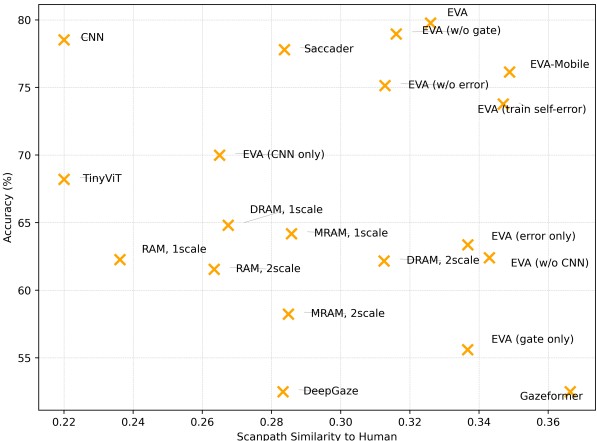

Figure 4: Comparison of hard-attention models on CIFAR-10 in terms of classification accuracy and composite scanpath similarity (SS).

### 4.3 PCA VISUALIZATION OF HIDDEN STATE DYNAMICS

To probe how sequential glimpses and multilayer recurrent processing evolve over time and across classes, we applied Principal Component Analysis (PCA) to the hidden states $h_t^1$ (lower RNN layer) and $h_t^2$ (higher layer) collected at each time step $t = 1, \ldots, T$. We visualized (i) time-step projections in PC-space to track how representations evolve through glimpses, and (ii) label-wise trajectories to inspect class separation. The higher-layer trajectories in Fig. 5 diverge strongly by class along PC2 as glimpse step $t$ increases along PC1, indicating progressive evidence accumulation and increasingly disentangled class representations. In contrast, the lower-layer states exhibit richer, more locally modulated dynamics, driven by prediction-error modulation and gate (compared to MRAM in Fig. Supp.8). These PCA results provide an interpretable link between EVA's internal dynamics, its gaze behaviour, and its final classification decisions.

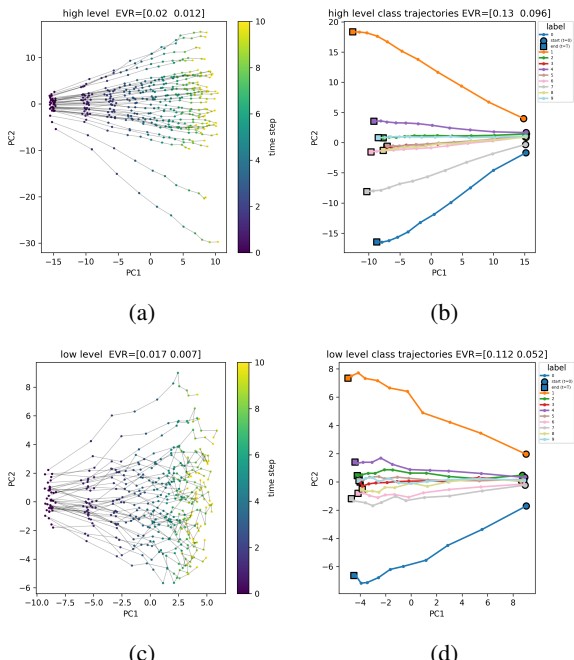

Figure 5: PCA of recurrent states in EVA on CIFAR-10. **Left**: trajectories of hidden states for 64 random test images, colored by glimpse step. Both the high-level (a) and low-level (c) RNNs show a consistent temporal progression, but the high-level state converges to a low-dimensional manifold. **Right**: class-wise mean trajectories obtained by averaging hidden states over images from the same class. High-level states (b) form well-separated, nearly 1-D class trajectories, whereas low-level states (d) remain more entangled. This supports our interpretation that the neocortical RNN implements an abstract, class-specific representation, while the SC-like RNN encodes more local, gaze-related dynamics.

## 5 DISCUSSION AND CONCLUSION

In this work, we proposed **EVA**, a lightweight brain-inspired hard-attention framework for modeling human active vision. EVA integrates three biologically motivated components: (i) a foveal CNN, (ii) a neuromodulatory controller that adjusts saccadic variability based on prediction error, and (iii) a pulvinar-inspired gate that regulates information flow between recurrent layers. To our knowledge, this is the first hard-attention model systematically evaluated against human gaze data. On CIFAR-10, EVA attains competitive accuracy with strong CNN and ViT baselines while producing emergent scanpaths that closely align with human eye movements. These results highlight the potential of EVA as both an efficient visual recognition model and a step toward interpretable AI systems that communicate through shared attention.

Taken together, our experiments reveal **three main insights**. First, EVA improves the trade-off between accuracy and efficiency among hard-attention models, while remaining competitive with strong CNN and ViT baselines. Second, the gate and neuromodulatory error modules make the learned scanpath policy more dynamic, task-aligned, and robust to gaze perturbations, even when they do not always increase accuracy. Third, EVA bridges performance and interpretability by producing human-like gaze patterns despite never seeing gaze labels. Overall, our findings suggest that scanpath similarity complements rather than replaces accuracy: integration of our brain-inspired modules yields models that are both powerful and behaviorally aligned with humans. **Limitations and future work.** EVA still suffers from training instability, calling for stronger reinforcement learning algorithms and better variance-reduction techniques. Our modules are only loosely inspired by neuroscience and are not intended as biologically faithful models. Finally, the gaze dataset (Gaze-CIFAR-10) is small and constrained compared with naturalistic settings, which may bias recorded scanpaths. Future work will stabilize training, extend EVA to broader tasks and richer gaze datasets, and design metrics that more faithfully capture human–model alignment.

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

APPENDIX

## A  THE USE OF LLMs

Portions of this manuscript were prepared with the assistance of large language models (LLMs). Specifically, LLMs were used to improve the clarity of English writing, polish grammar, and suggest alternative phrasings for the paper. All scientific ideas, experimental designs, implementations, and analyses were conceived and carried out by the authors. The authors take full responsibility for the content of the paper, including the correctness of technical claims and the reported results.

## B  REPRODUCIBILITY STATEMENT

We have taken several steps to ensure the reproducibility of our work. All datasets used in this study (CIFAR-10, ImageNet-10, and Gaze-CIFAR-10) are publicly available; detailed preprocessing steps are described in Appendix C. The full model architecture, including the CNN foveal module, neuromodulator, and pulvinar gate, is specified in Section 3, with implementation details and hyperparameters provided in Appendix D. The baseline models results are based on public available deposits such as Yoshioka (2024). Training procedures, including optimization settings, ablation configurations, and evaluation metrics (accuracy, FLOPs, DTW, ScanMatch, NSS, AUC), are described in Section 4. Additional analyses of training stability, gaze perturbation, and qualitative scanpath visualization are also included in the supplementary material. To facilitate reproduction of our results, we will release the source code in Github after the double blind processing. Together, these resources are intended to make it straightforward for others to reproduce and extend our findings.

## C  SCALABILITY CHECK TO DIVERSE DATASETS AND TASKS

### C.1  HIGH RESOLUTION IMAGES

On the image classification subset ImageNet-10, which requires higher-resolution natural images, our model EVA scaled favourably compared to both convolutional and hard-attention baselines (Table Supp.1).Saccader-Mobile adopts the Saccader architecture of Elsayed et al. (2019). but replaces the original BagNet-77 backbone with MobileNet-V3. Importantly, Saccader first applies the CNN densely to patches or glimpses of the entire image to obtain logits at all spatial locations, and the attention module then selects a sequence of locations whose logits are averaged to form the final prediction. Thus, although only a fraction of the image is computed through CNN like EVA, the computation still iterate over the whole image, and result in high computation demand. In contrast, EVA-Mobile only feeds a small number of high-resolution foveal glimpses only through the backbone, a low-resolution peripheral crop is only processed by the fully connected layer. The classifier operates on a compressed fovea–periphery representation produced by an RNN, without access to a dense grid of logits over the whole image. EVA therefore uses strictly less visual information than Saccader and achieves lower FLOPs (8.87B vs. 37.27B). As a consequence, Saccader-Mobile reaches slightly higher ImageNet accuracy (75.9% top-1, 94.5% top-5) than EVA-Mobile (71.92% / 91.92%), but at 4.2× higher computational cost. We therefore treat Saccader as a strong hard attention baseline: it can be viewed as a full-image CNN or the modern transformer models with an interpretable readout, whereas EVA enforces a much stricter foveation constraint, and produce more human-like sequential gaze behaviour, especially in temporal order, as shown in the CIFAR-10 and COCO-search experiment.

Table Supp.1: Results on ImageNet-100. (pre.) indicates initialization from ImageNet-1K pretrained weights; (scratch) indicates random initialization. All hard-attention baselines without CNN are trained from scratch. The CNN backbone is MobileNetV3 when -Mobile is indicated, otherwise the module is same as described in Fig. Supp.2

| Model | FLOPs (B) | Top-1 Acc. (%) | Top-5 Acc. (%) |
|---|---|---|---|
| MobileNet (scratch) | 2.2 | 47.64 | 76.12 |
| MobileNet (pre.) | 2.2 | 80.36 | 96.5 |
| Saccader-Mobile (pre.) | 37.27 | **75.9** | **94.5** |
| RAM | 0.35 | 11.26 | 32.78 |
| DRAM-Mobile (pre.) | 8.83 | 19.00 | 43.12 |
| DRAM | 1.06 | 21.44 | 47.3 |
| MRAM | 0.35 | 12.88 | 34.76 |
| EVA-Mobile (scratch) | 8.87 | 42.62 | 68.24 |
| EVA-Mobile (pre.) | 8.87 | 71.92 | 91.92 |
| EVA | **6.6** | 65.86 | 80.24 |

Table Supp.2: COCO-search18: scanpath metrics.

| Model | COCO Acc.% ↑ | COCO-search Acc.% ↑ | DTW ↓ | SM ↑ | NSS ↑ | AUC ↑ |
|---|---|---|---|---|---|---|
| CNN MobileNet (pre.) | 58.82 | 27.5 | - | - | - | - |
| Gazeformer | - | - | 168.39 | 0.571 | 1.961 | 0.8 |
| Saccader-Mobile (pre.) | 57.1 | 16.7 | 333 | 0.242 | 0.361 | 0.658 |
| RAM | 34.81 | 12.83 | 500.01 | 0.072 | -0.132 | 0.585 |
| DRAM | 43.32 | 14.34 | 530.78 | 0.077 | -0.124 | 0.587 |
| MRAM | 35.86 | 14.17 | 624.57 | 0.015 | -0.09 | 0.605 |
| EVA-Mobile (scratch) | 45.81 | 14.79 | 513.48 | 0.101 | -0.074 | 0.593 |
| EVA-Mobile (pre.) | 55.82 | 16.63 | 280.29 | 0.313 | 0.307 | 0.714 |

## C.2 OBJECT DETECTION TASK

We also include a workload for classification in object detection task to explore how EVA generalises beyond classification. We emphasise that this is a scalability check rather than a full performance comparison. Because the detection task (e.g., on COCO images) involves recognising often small-object targets in cluttered scenes, the architecture of EVA (with a sequential glimpse RNN policy, especially the prediction error module contains an inherent bias for classification: all glimpse are localized parts from the same object) is less suited and strong feature encoding becomes dominant. Accordingly, EVA does not match the best specialized detectors, we include these results to show the limits of our current design and to motivate future extensions. In this experiment, we took figures that contains the 10 labels: (bottle, bowl, car, chair, clock, cup, keyboard, laptop, microwave, tv) from the COCO 2014 object detection dataset, and train it in a image recognition fashion: predict labels at the last time step. Even the classification accuracy is low as expected, and we hypothesize that using a pre-trained feature network, a MobileNetV3Large model with pretrained weight on ImageNet, can compensate partially the difficulty of training a hard attention model, and thus can provide insight on the hard attention model's performance.

From the Table Supp.2, the result has supported the EVA model can produce more human-like scanpath than other hard attention models while a performance gap between scanpath prediction models like Gazeformer. The result is acceptable because our model is never trained on human scanpath, and only utilize it for a content-based task. Unlike Saccader model, EVA model takes only patches of glimpsed images as input, reducing much computational cost, while achieving comparable performance, nevertheless producing scanpath that is closer to human.

## C.3 QUALITATIVE RESULTS

From the visualization of EVA model in ImageNet in Fig. Supp.1, the EVA model learned to focus on the object while being actively exploring with several glimpses. The result shows the scalability of emerging human-like scanpath from low resolution CIFAR-10 image to real-world image classification.

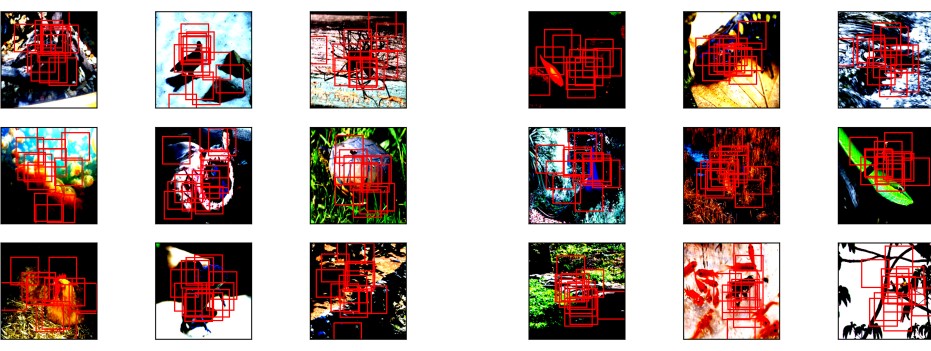

(a) EVA at 193 epoch                    (b) EVA-Mobile (pre.) at 47 epoch

Figure Supp.1: Visualization of EVA in ImageNet100, Red rectangular is the boundary of foveal vision, peripheral vision is hided for better visualization. Animations of dynamic scanpath visualizations at each time and code are in the anonymous repository: https://anonymous.4open.science/r/Anon-EVA-8607/

# D DETAILS OF EXPERIMENTAL SETTING

In the main classification experiments, we use CNN baselines without ImageNet pretraining to ensure a fair comparison with hard-attention models. In contrast, some hard-attention models, including Saccader and EVA-Mobile, use pretrained ResNet or MobileNet CNN modules (trained as our baselines) as backbones. Specifically, instead of BAGNet, we use ResNet in Saccader for fairness of comparison. In EVA, we use a simple CNN module described in Fig. Supp.2. The DRAM 1-scale and 2-scale variants use the same CNN architecture as in EVA. Because Saccader requires a pretrained CNN, we did not test Saccader with the simple CNN design in EVA. For Saccader, patches are selected on a $8 \times 8$ discrete grid and ordered by score to form a pseudo-scanpath for visualization.

The ViT baseline uses a patch size of 4, hidden dimension 512 with 4 layers, 6 heads, MLP dimension 256, and dropout 0.1. For all hard-attention models, including RAM, DRAM, MRAM, and EVA, we fix the hyperparameters across experiments: 12 glimpse steps, two patches per step (a foveal crop of size $8 \times 8$ and a concatenated peripheral crop of size $16 \times 16$). If we indicate "1scale", only the foveal image is used and there is no peripheral image. FLOPs are calculated under the same condition of a batch of 9 images during testing. The FLOPs in Table 1 for hard-attention models are computed over the complete glimpse sequence (12 glimpses). The random seed is fixed to 1 in all experiments for reproducibility and fair comparison. The scanpath of hard-attention models in Table 1 are generated using CIFAR-10 image data, which is a standard 32x32 size. Scanpaths generated by scanpath prediction models including Gazeformer and DeepGaze model uses the 1024x1024 Gaze-CIFAR-10 image data. Both coordinated are normalized to 224x224 for computation with human scanpath coordinates. We report the hard-attention models' scanpath results using 1024x1024 Gaze-CIFAR-10 image data with SS metrics in the Table Supp.3 as part of the robustness test.

## D.1 DETAILS OFSCANPATH SIMILARITY METRICS

For the composite scanpath similarity in Eq. equation 20, the metrics $\mathcal{D}, \mathcal{S}, \mathcal{N}, \mathcal{A}$ are normalized using standard linear scaling. Let $m$ denote the raw metric value for a given model and $m_{\min}, m_{\max}$ be the minimum and maximum values. Specifically, one can naturally think out to use human data itself as target, so we calculated the ideal upper bound of the scanpath metrics with the exact

same human-scanpath. The ideal upper bound of metrics is $\mathcal{D}_{max} = 0.003, \mathcal{S}_{max} = 1.0, \mathcal{N}_{max} = 6.052, \mathcal{A}_{max} = 0.995$. However, the lower bound of scanpath is non-trivial, because it's controversial on the definition of whether a scanpath is bad. In simplification, we used a pseudo-scanpath that fixes gazes at the corner of the image, assuming the visual scanpath is bad because it always focusing on the off-target region and losses the temporal feature. In this case, we obtained the lower bound that applied in this experiment being $\mathcal{D}_{min} = 2023.87, \mathcal{S}_{min} = 0.013, \mathcal{N}_{min} = -0.053, \mathcal{A}_{min} = 0.541$. We are also confirmed that these metrics are scanpath-based, because if we are fixing the gazes at the center of image, the metrics didn't improve much: $\mathcal{D}_{min} = 2040.0, \mathcal{S}_{min} = 0.013, \mathcal{N}_{min} = -0.052, \mathcal{A}_{min} = 0.541$.

For metrics where larger is better (ScanMatch, NSS, AUC), we use

$$m' = \frac{m - m_{\min}}{m_{\max} - m_{\min}}, \tag{3}$$

and for metrics where smaller is better (DTW), we use

$$m' = \frac{m_{\max} - m}{m_{\max} - m_{\min}}. \tag{4}$$

Thus all normalized scores $m' \in [0, 1]$ have the same "higher is better" direction; in Eq. equation 20 we set $\mathcal{D} = m'_D, \mathcal{S} = m'_{SM}, \mathcal{N} = m'_{NSS}$, and $\mathcal{A} = m'_{AUC}$.

## D.2 DETAILS OF STABILITY TEST

To assess the stability of EVA under gaze perturbations, we evaluate the model under several controlled "lesioned" gaze policies, detailed here for completeness.

**Center-fixed** forces all glimpses to the image center at every time step, removing any learned spatial exploration on glimpse and testing how much performance can be supported by a pure central bias. While image classification benchmarks can be heavily center-biased, the drop of performance in EVA suggesting potential the model is the hard attention model that the least biased on the center or local-based information.

**Corner-fixed** fixes the glimpse location deterministically through the one of the image corners, in this experiment, the glimpse is fixed at the top-right corner, regardless of the input, probing robustness when attention is systematically misaligned with the object of interest.

**Random** sampling draws each glimpse location independently from a uniform distribution over the image plane at every step, preserving the number of glimpses but discarding all learned spatial structure with glimpse. DRAM is low performance as expected, because its global features CNN might be biased, restraining it from learning features uniformly distributed over the image. Oppositely, EVA performs well indicating its learn features are not biased on any short-cut or local region.

Finally, **Shuffled** uses EVA's own predicted locations but randomly permutes their order along the time axis, thereby preserving the exact set of attended points while destroying the temporal sequence in which they are visited. Intriguingly, EVA losses most accuracy among hard attention models suggesting the high reliance on temporal structure in saccade akin to **visual reasoning**. For example, the order of viewing from ocean background to an airplane object and the order of visiting from the airplane object to the ocean background have a significant influence on the classifier's prediction as illustrated in Figure Supp.11b. It is also interpretable by human because we can expect a ship from the ocean background, while when the airplane is first observed, the prediction will be almost certain unless new evidence appears. Together, these perturbations in Table 2 disentangle different contributions of the learned policy—central bias, spatial selection, and temporal strategy, and allow us to quantify how much EVA's performance depends on each component.

# E   METHOD FOR SEQUENTIAL FIXATIONS EXTRACTION AND ALIGNMENT OF CIFAR-10 AND GAZE-CIFAR-10 IMAGES

In this chapter, we describe the methodology used to establish a reliable mapping between images from the original CIFAR-10 dataset and the corresponding human gaze data, which was recorded using upsampled images at a resolution of 1,024×1,024 pixels.

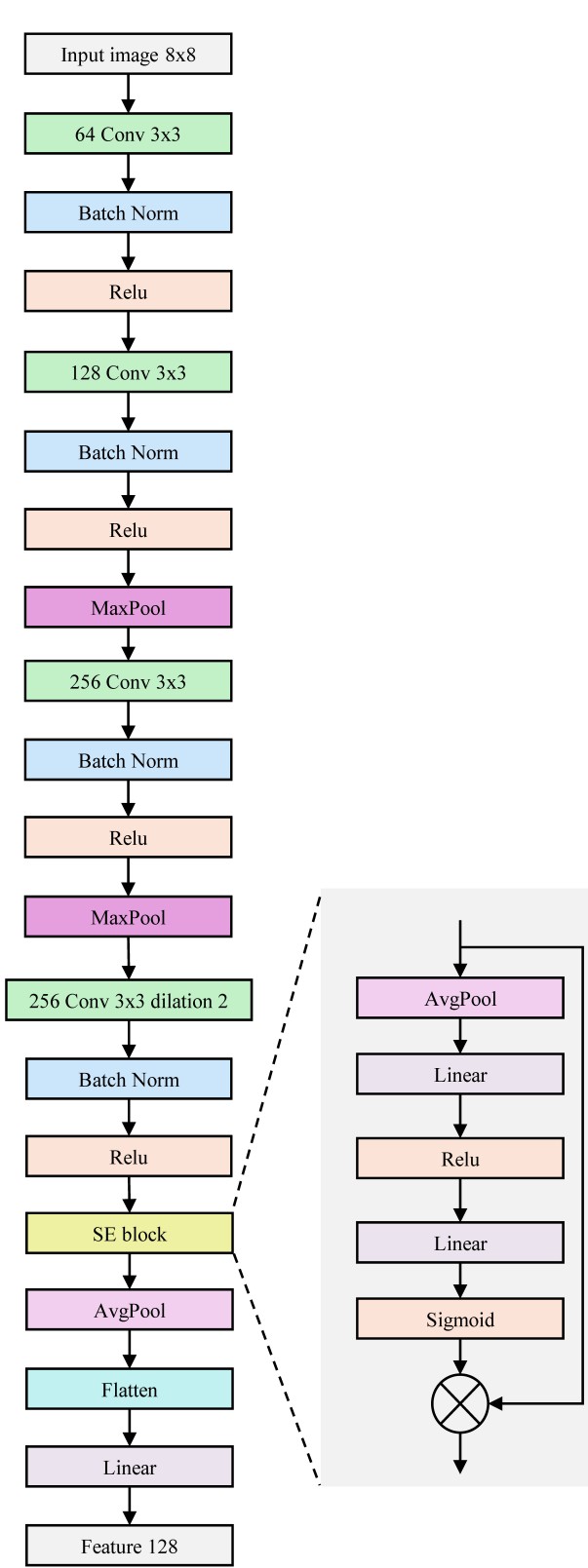

Figure Supp.2: CNN module used in Bravo. In Bravo-large, the lightweight CNN is increased size from 64-128-256-256 to 96-192-384-384, without changing other parts in the architecture.

## E.1    IMAGE ALIGNMENT

The human gaze data we utilized was recorded on images obtained by upsampling the original 32×32 CIFAR-10 images to a much higher resolution of 1,024×1,024 pixels with Real-ESRGAN. Since the Gaze-CIFAR-10 dataset did not explicitly provide a direct mapping between the original CIFAR-10 images and their corresponding upscaled counterparts, we implemented an image-hashing-based matching technique to establish precise correspondence.

To achieve robust alignment, we adopted a perceptual hashing approach (pHash), which effectively captures image content irrespective of resolution changes, slight color variations, or compression artifacts. The procedure involved the following steps:

- Construction of Reference Hash Database: We first combined all 10,000 CIFAR-10 images from test split into a single reference set. For each original CIFAR-10 image, we computed its pHash, creating a dictionary mapping each hash to the original image index and corresponding CIFAR-10 class label.

- Matching Procedure: For each upscaled human gaze image, we downsampled it back to the original CIFAR-10 resolution (32×32) using bilinear interpolation. We then computed the pHash of this downscaled image and looked it up within our precomputed hash database. Exact matches provided immediate identification. If no exact match occurred, the nearest perceptual hash with minimal Hamming distance was used to ensure robust correspondence despite minor differences introduced during the upsampling and subsequent downsampling processes.

- Quick Correspondence Check: The hashing-based matching allowed us to reliably recover the original CIFAR-10 image indices, and labels corresponding to each human gaze recording. To ensure precise match, we run a random sample check by showing 20 samples per label of CIFAR-10 image and manually check the correspondence between original and upsampled images.

## E.2    SEQUENTIAL FIXATIONS EXTRACTING

The raw gaze recordings typically containing hundreds of gaze points coordinates per image are normalized to the range [0, 224]. After aligning images, the raw gaze recordings were clustered spatially and temporally using an I-DT (Identification by Dispersion Threshold) approach. Specifically, we grouped gaze points that fell within a defined spatial radius (15 pixel distance) into fixation clusters, computed the center of each fixation cluster, and finally retained exactly 12 sequential fixation clusters per gaze recording, aligning with the number of fixations predicted by our model.

With image correspondence and gaze clustering complete, we obtained consistent and comparable human scanpaths with 12 fixation points per image aligned in space and sequence with model-predicted gaze data. This alignment enabled rigorous quantitative evaluation of our model's performance in terms of scanpath similarity metrics, thereby providing robust and interpretable results.

## F    SENSITIVITY CHECK OF HYPERPARAMETERS IN EVA

In EVA, we introduces multiple new hyperparameters to hard attentions, and in this chapter, we investigated the sensitivity of these hyperparameters. Due to the scale of number of hyperparameters, even it usually takes 100-200 epochs to reach optimal performance, we compare the performance of EVA models at epoch 20 on CIFAR-10 here. The hyperparameters mainly distributed in the neuromodulator that controlling the sigma with prediction error, and the initial value of the learnable decaying parameter. In the neuromodulator, there are 4 parameters in total: $\tau_{long}$, $\tau_{short}$ is the decay factor in Eq.equation 6, $\sigma_{\min}$, $\sigma_{\max}$, in Eq.equation 8. In the pulvinar gate, there is only one decaying parameter $\gamma$ in Eq.equation 10. Fig. Supp.3 summarizes the hyperparameters sensitivities. From the result, we see the $\sigma_{\min}$ has the largest influence to the performance of EVA model, while the decay is a learnable parameter that has the least influence. The result aligns with the intuition that the hard attention model EVA can learn better with a larger $\sigma_{\min}$, that the model has to constantly exploring more to gather information, and the optimal value of $\sigma_{\min}$ is found at 0.5. As for $\tau_{long}$, $\tau_{short}$, our results shows as long as the difference between them is large enough, the $\tau_{long}$, $\tau_{short}$ can guarantee

good performance. Notably, in the main experiment, we used 0.05 as the $\sigma_{\min}$, and the accuracy was 79.77%, but when we finished the training of the optimal hyperparameters here, and the accuracy can further grows to 82.51% on CIFAR-10.

Additionally, we tested the influence of number of glimpse to EVA, and the result is shown in Fig. Supp.10. We showed with only 2 glimpses (at least 2 glimpses are need the calculation for neuromodulator), EVA can have approximately 62% accuracy in only 2 glimpses, and can further have over 75% accuracy in 5 steps.

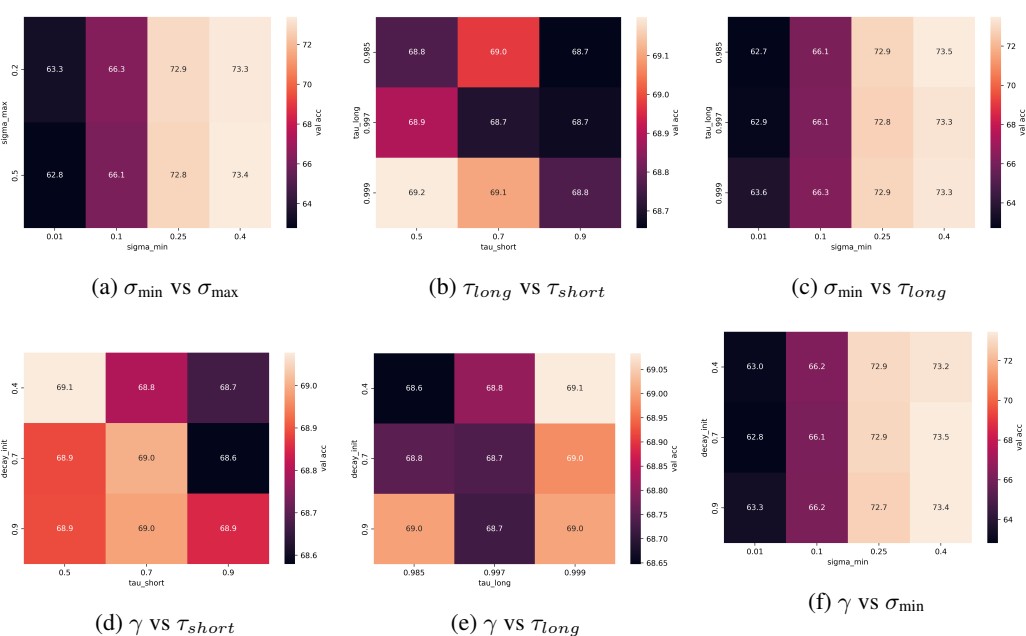

(a) $\sigma_{\min}$ vs $\sigma_{\max}$      (b) $\tau_{long}$ vs $\tau_{short}$      (c) $\sigma_{\min}$ vs $\tau_{long}$

(d) $\gamma$ vs $\tau_{short}$      (e) $\gamma$ vs $\tau_{long}$      (f) $\gamma$ vs $\sigma_{\min}$

Figure Supp.3: Hyperparameter sensitivity check of EVA on CIFAR-10.

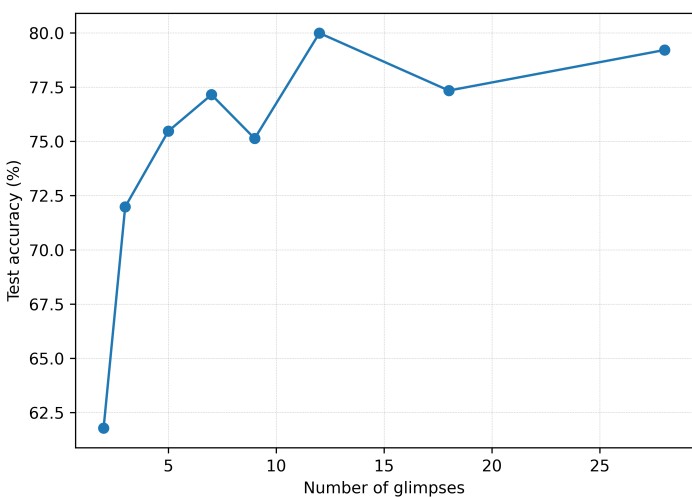

Figure Supp.4: Comparison of number of glimpse to EVA model accuracy.

# G ADDITION PCA ANALYSIS OF HIDDEN STATE DYNAMICS RESULTS

From the PCA visualisation of hidden states over glimpse steps, RAM exhibits a characteristic "radiation–like" pattern in both sample-wise and class-wise plots. PCA extracts the leading directions of variance, PC1 explains the largest share, followed by PC2. In RAM, PC1 is predominantly label–related, while PC2 correlates more strongly with glimpse step (time). The label–wise embeddings show that trajectories quickly diverge by class along PC1 and then change only weakly as glimpses proceed, indicating that the RNN state saturates early and later glimpses bring limited additional discriminative structure. This behaviour is consistent with the original RAM results (Mnih et al., 2014), where performance peaks at an intermediate number of glimpses and degrades when the sequence is made shorter or longer.

In DRAM (Ba et al., 2014), high-level RNN, the first two PCs show almost identical trajectories across classes, suggesting that the dominant recurrent dynamics encode generic context rather than class-specific information. This is consistent with the design of DRAM, where a full-image context vector provides a rich global summary upfront, and the RNN's temporal evolution plays a more limited role in building discriminative features. We further observe that trajectories in the high-level DRAM RNN resemble the low-dimensional manifolds reported for self-organized robot motor control in (Han et al., 2020), whereas the lower-level DRAM RNN behaves more like other hard-attention models high-level RNN (MRAM, EVA). The lower laye in DRAM is mainly responsible for classification, with its first PCs aligned with glimpse-step evolution.

Finally, the 2–scale MRAM (foveal + peripheral input) yields a higher-layer RNN whose class-wise trajectories are more stable and clearly separated than in the 1-scale variant, even though its classification accuracy is slightly lower. This suggests that the human-inspired retinal design primarily reshapes the recurrent state space and gaze behaviour toward more human-like, interpretable dynamics rather than simply maximizing accuracy.

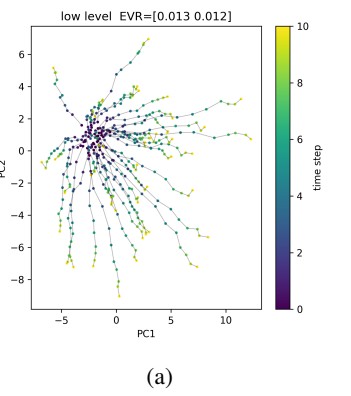

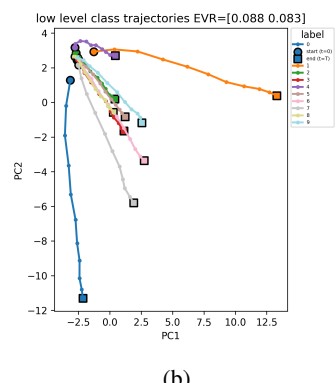

(a)                                          (b)

Figure Supp.5: PCA visualization of RAM, time-step projection (left) and label-wise embedding (right).

## G.1 ROBUSTNESS ON CIFAR-10 AND ALIGNMENT OF EVA SCANPATH IN GAZE-CIFAR-10 IMAGES

We performed three tests as evaluating the robustness of EVA against standard baselines, a projected gradient descent (PGD) test, a random occlusion (RO) test, and a zero-shot testing on the downsampled Gaze-CIFAR-10 Images from $1024 \times 1024$ to $32 \times 32$, and recollected the scanpath of each models. PGD is a standard adversarial test of robustness of a vision model, and in RO test, we randomly occlude several regions of images. We set the perturbation $\epsilon = 8/255$ with 20 steps in PGD test, and 5 occlusion patches size at $16 \times 16$. Tabel Supp.3 shows the results. From the result, we see non-CNN backed or light CNN backed hard attention models has the best resistance to PGD attack. ViT and EVA models have the best performance in RO test and Gaze-CIFAR test. Notably, while there isn't much change in the scanpath similarity metrics of hard attention baselines, EVA gained more

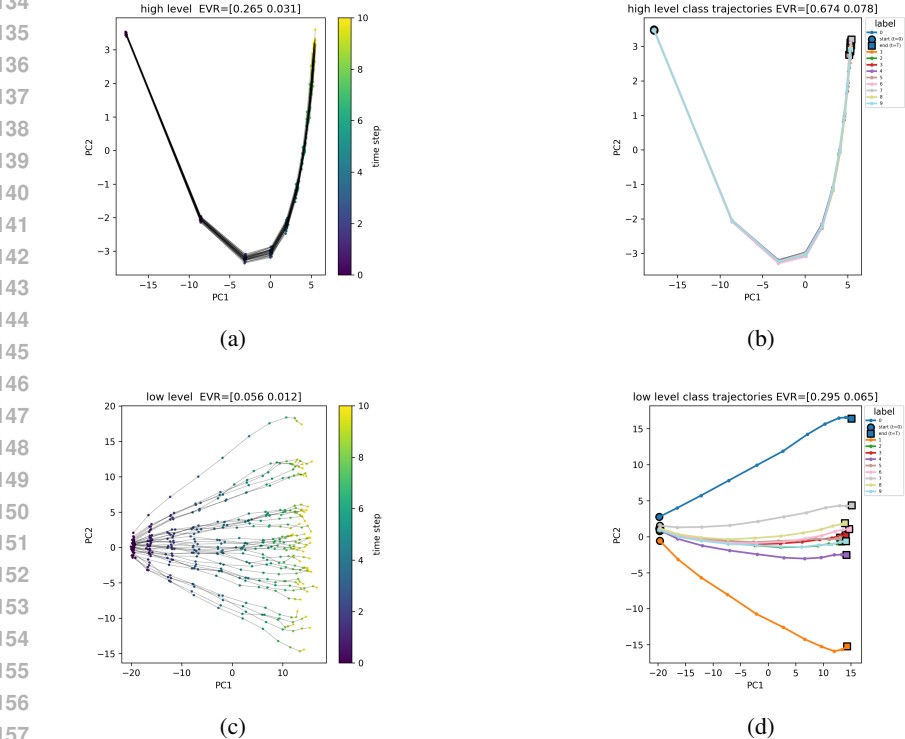

Figure Supp.6: PCA visualization of DRAM, time-step projection (left) and label-wise embedding (right).

scanpath similarity in the Gaze-CIFAR image, with more test accuracy, supporting the robustness of EVA, and the emerged-scanpath align better on the same image where human gaze is taken.

Table Supp.3: Robustness of various models on CIFAR-10 under two corruption protocols, and on Gaze-CIFAR-10 images

| Model | PGD Acc. | RO Acc. | Gaze-CIFAR Acc. | DTW ↓ | SM ↑ | NSS ↑ | AUC ↑ |
|---|---|---|---|---|---|---|---|
| ResNet18 | 0.42% | 36.47% | 39.79% | - | - | - | - |
| MobileNetV3 | 0.36% | 31.33% | 45.41% | - | - | - | - |
| ViT | 0.07% | 46.12% | 63.20% | - | - | - | - |
| Saccader | 17.21% | 43.80% | 33.37% | 918.12 | 0.274 | 0.278 | 0.666 |
| RAM, 1scale | 20.75% | 27.60% | 42.89% | 1118.95 | 0.253 | 0.302 | 0.650 |
| RAM, 2scale | 20.28% | 27.03% | 45.69% | 1169.29 | 0.259 | 0.372 | 0.683 |
| DRAM, 1scale | 20.16% | 29.82% | 46.15% | 1036.01 | 0.261 | 0.277 | 0.657 |
| DRAM, 2scale | 19.18% | 28.53% | 46.44% | 823.79 | 0.307 | 0.665 | 0.678 |
| DRAM-ResNet | 19.36% | 25.46% | 44.37% | 788.29 | 0.303 | 0.465 | 0.689 |
| MRAM, 1scale | 18.35% | 21.78% | 44.70% | 871.82 | 0.292 | 0.388 | 0.676 |
| MRAM, 2scale | 17.96% | 24.46% | 45.07% | 942.80 | 0.262 | 0.326 | 0.672 |
| EVA | 17.80% | 45.86% | 53.51% | 797.29 | 0.335 | 0.612 | 0.702 |
| EVA-Mobile | 16.38% | 25.25% | 51.21% | 761.11 | 0.339 | 0.728 | 0.707 |

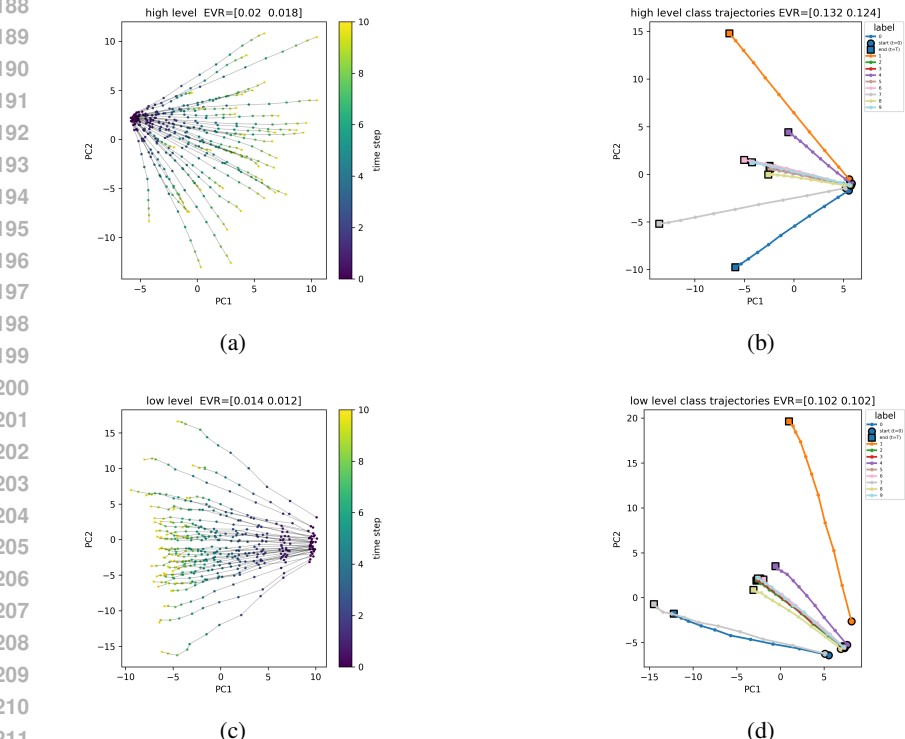

Figure Supp.7: PCA visualization of MRAM 1 scale, time-step projection (left) and label-wise embedding (right).

**Sensitivity of Brain-inspired Modules in EVA**  In the module-level design, as the purpose of this study is to utilize brain-inspired architectures for hard attention models. We modeled the thalamic (pulvinar) gate, but it's not a simple gate simulating only the pulvinar. From the study of neuroscience, the recent study shows thalamus has much relation to the rhythmic adjustment that the top-down signal and bottom-up signal are traveled in different layers of cortex and in different frequencies, and the recurrent structures broadly exist especially in neocortex and prefrontal cortex (Miller et al., 2018). This indicate the need of recurrence in Eq.equation 15, but empirically, we found by concatenating them, there will be increase of hyperparameters that introducing unexpected training cost. So we did comparative experiments of EVA on the CIFAR-100, with three setting of recurrence: by simply adding the hidden state $\rho_t$ and the $\mathbf{z}_t$ (AR); concatenating $\mathbf{z}_t \parallel \rho_t$ (CR); and no recurrence (NR) as baseline. There are same dropout to $\rho_t$ which simulate the sparse connections in cortex. The baseline model of no recurrence get 50.30% accuracy, the concatenating get 48.31%, and the simple addition case get 49.21%. Fig. Supp.10 further shows the difference of prediction by these different configurations. This result shows recurrence did introduce extra cost to training, while simply adding can retain relatively high accuracy.

The difference in CNN architecture is tested by comparing in larger CNN and the MobileNetV3 backbone. We tested EVA with MobileNetV3 with pretrained weights on ImageNet, the model can reach 76.14% accuracy while no pretrained weight model can get 74.96% accuracy. The influence of CNN module is not large as long as it's well designed for the low resolution images at $8 \times 8$ size.

# H  DATA ENHANCEMENT ON CIFAR-10

Data enhancement is an efficient method for preventing the model from over-fitting. By applying the RandomCrop and RandomHorizontalFlip for data enhancement, models often gain more accuracy in generalizing to testing. Even though these enhancements are not applied in this work for fairness comparison, we tested small samples of our models and CNN and ViT baselines in tables Supp.4 to show

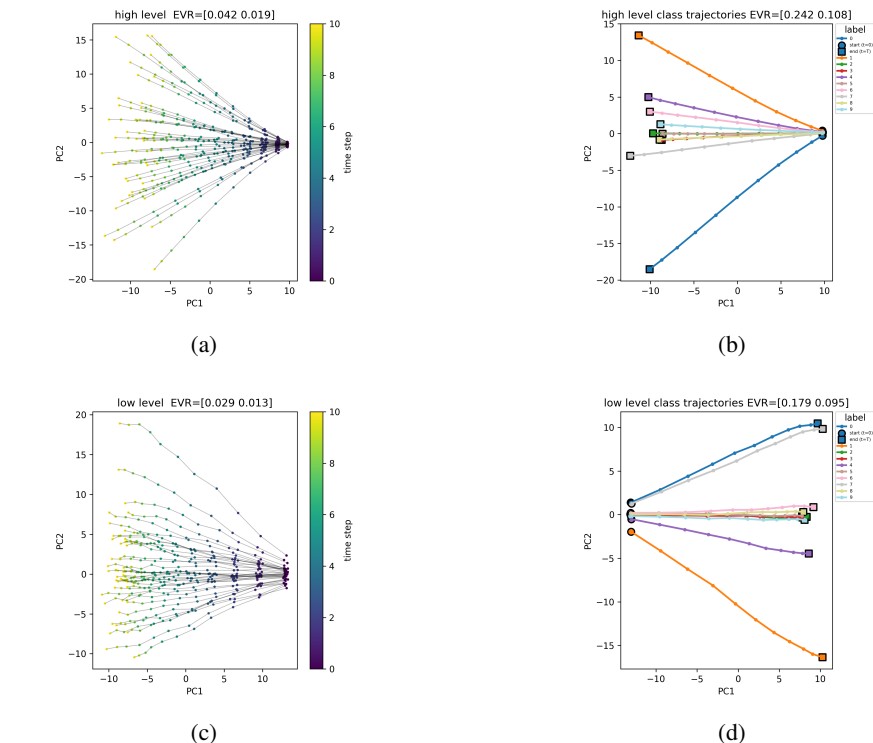

Figure Supp.8: PCA visualization of MRAM 2 scale, time-step projection (left) and label-wise embedding (right).

the potential scaling difference between our model and baselines with these techniques. The uniform improvements indicate that the pipeline enriches the diversity of the training distribution rather than catering to a specific inductive bias. In particular, the gains on ResNet18 and MobileNetV3 imply that standard convolutional features already profit from the added texture and geometric variations. For ViT, the method acts as an implicit regulariser, alleviating over-fitting on the comparatively tiny CIFAR-10 images. Notably, EVA models are less sensitive to these data enhancement techniques, because it reached optimal performance with limited data.

Table Supp.4: Data enhancement results on CIFAR-10

| Model | Params (M) | Infer Time (ms/im) | Accuracy | Accuracy Gain |
|---|---|---|---|---|
| CNN (ResNet18) | 11.18 | 0.91 ± 0.03 | 85.99% | 7.99% ↑ |
| CNN (MobileNetV3) | 4.21 | 0.90 ± 0.05 | 88.34% | 9.82% ↑ |
| ViT-tiny | 4.37 | 2.98 ± 0.26 | 79.81% | 11.6% ↑ |
| EVA | 4.21 | 2.92 ± 0.17 | 84.02% | 5.09% ↑ |

# I EXAMPLES FROM THE EVA MODEL

We present additional examples of predicted scanpath by EVA model here, where predictions at each glimpse step is visualized. EVA is more stable during the visual processing simulating human saccades, yet loyalty to the new prediction if evidence is found somewhere else. From visualization of visible region develops as the glimpse moves in time, in Figure Supp.11, we present the interpretability of the EVA model: the model made false prediction based on the background, for example, predicting a ship on a ocean background, and mistaken deer and horse, and the prediction becomes confident and stable when objects are better explored.

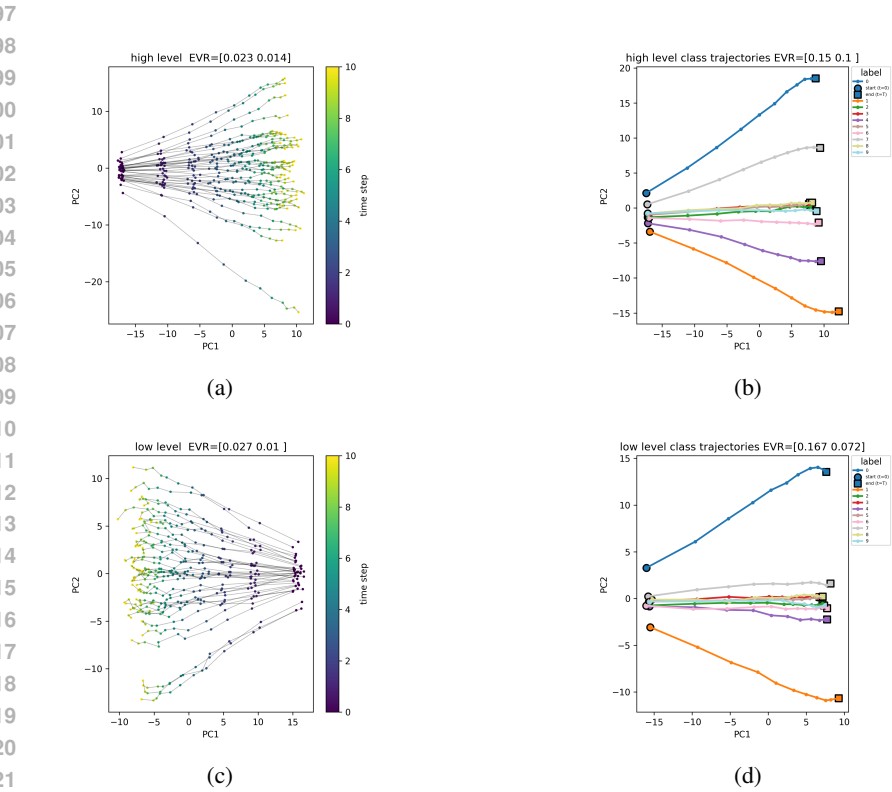

(a)                                        (b)

(c)                                        (d)

Figure Supp.9: PCA visualization of EVA-Mobile, time-step projection (left) and label-wise embedding (right).

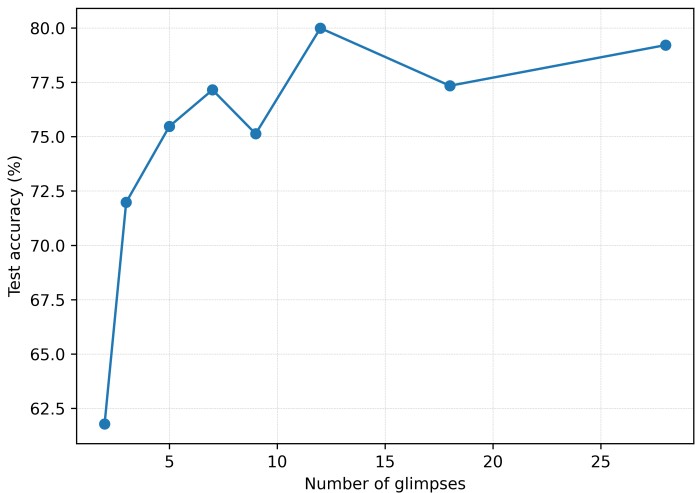

Figure Supp.10: Comparison of number of glimpse to EVA model accuracy.

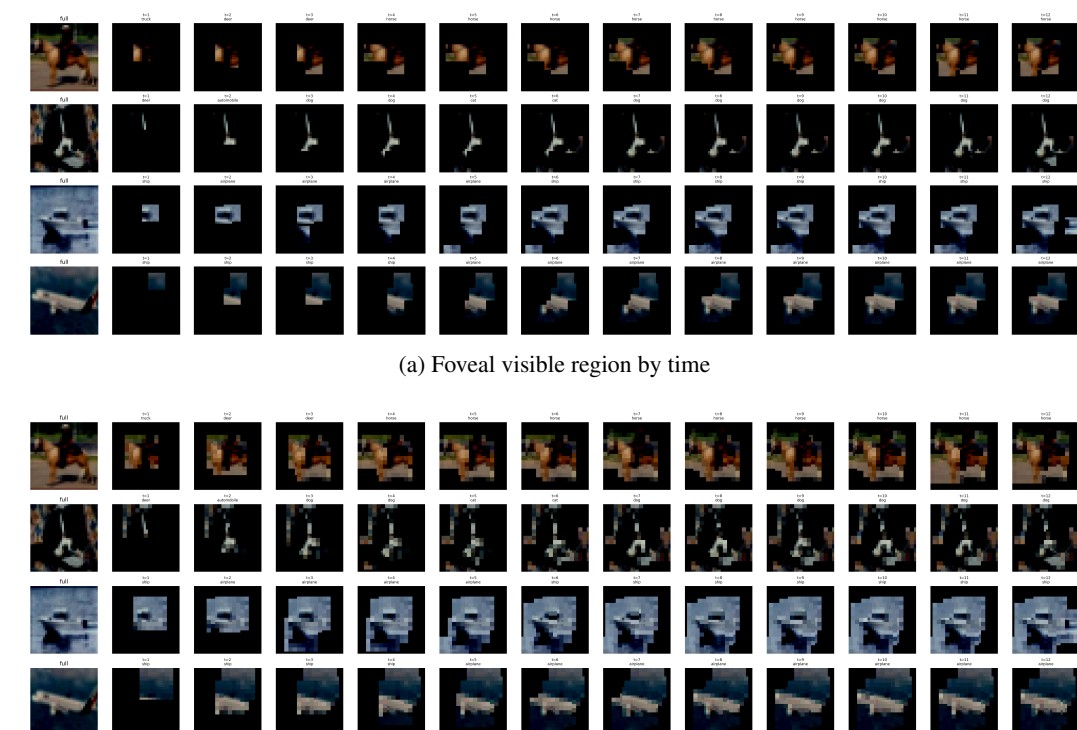

(a) Foveal visible region by time

(b) Peripheral visible region by time

Figure Supp.11: Sample 1 of visualization of EVA in CIFAR-10, with glimpse time t and prediction.

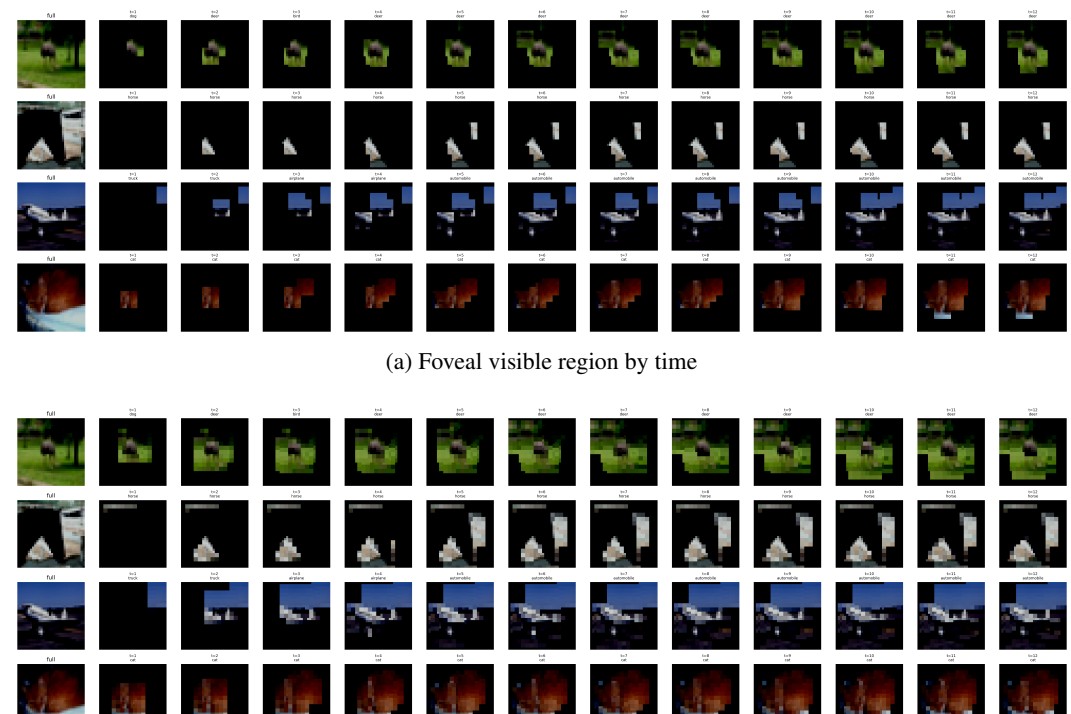

(a) Foveal visible region by time

(b) Peripheral visible region by time

Figure Supp.12: Sample 2 of visualization of EVA in CIFAR-10, with glimpse time t and prediction.

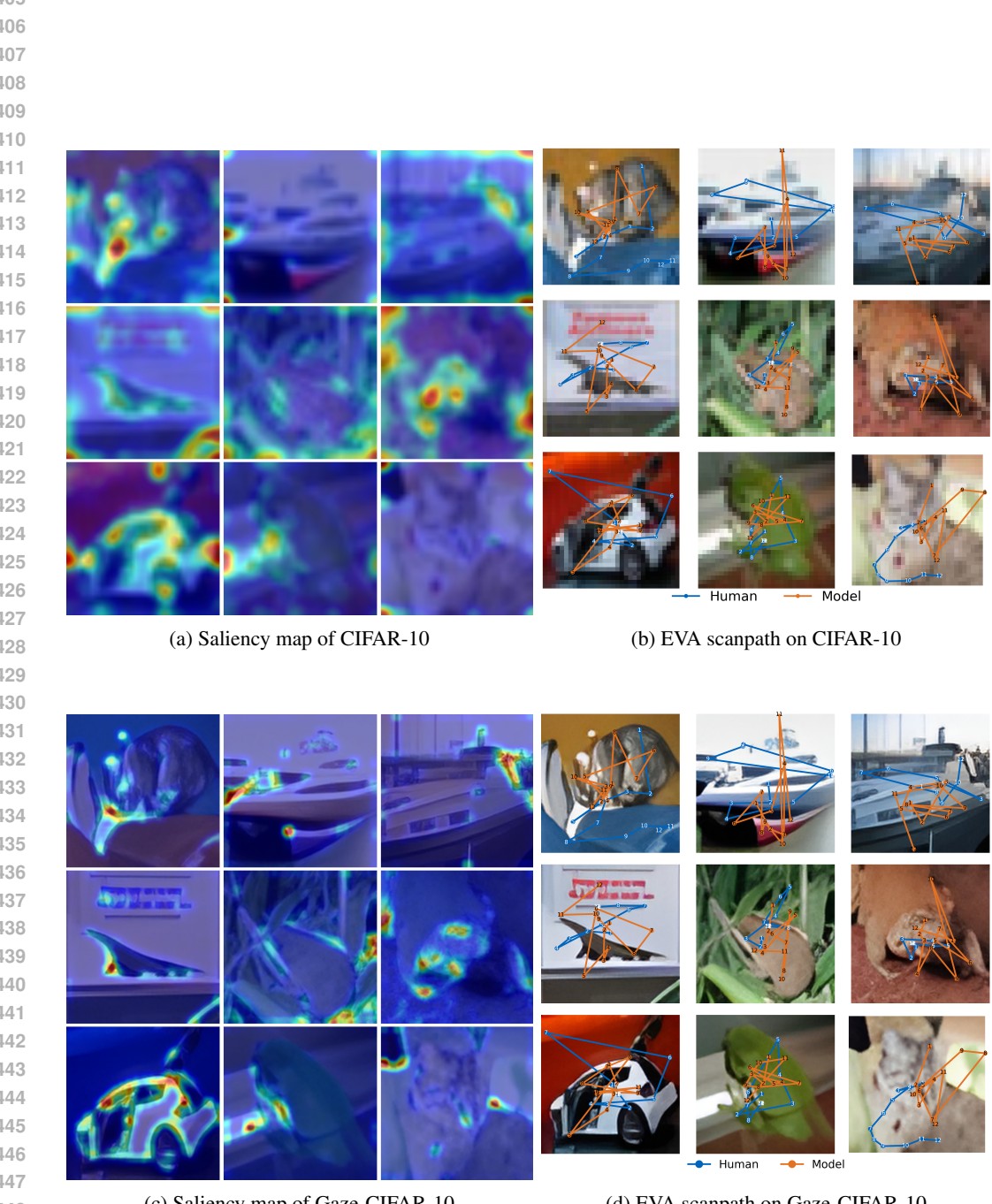

(a) Saliency map of CIFAR-10          (b) EVA scanpath on CIFAR-10

(c) Saliency map of Gaze-CIFAR-10      (d) EVA scanpath on Gaze-CIFAR-10

Figure Supp.13: Comparison of scanpath in CIFAR-10 and Gaze-CIFAR-10 with saliency.

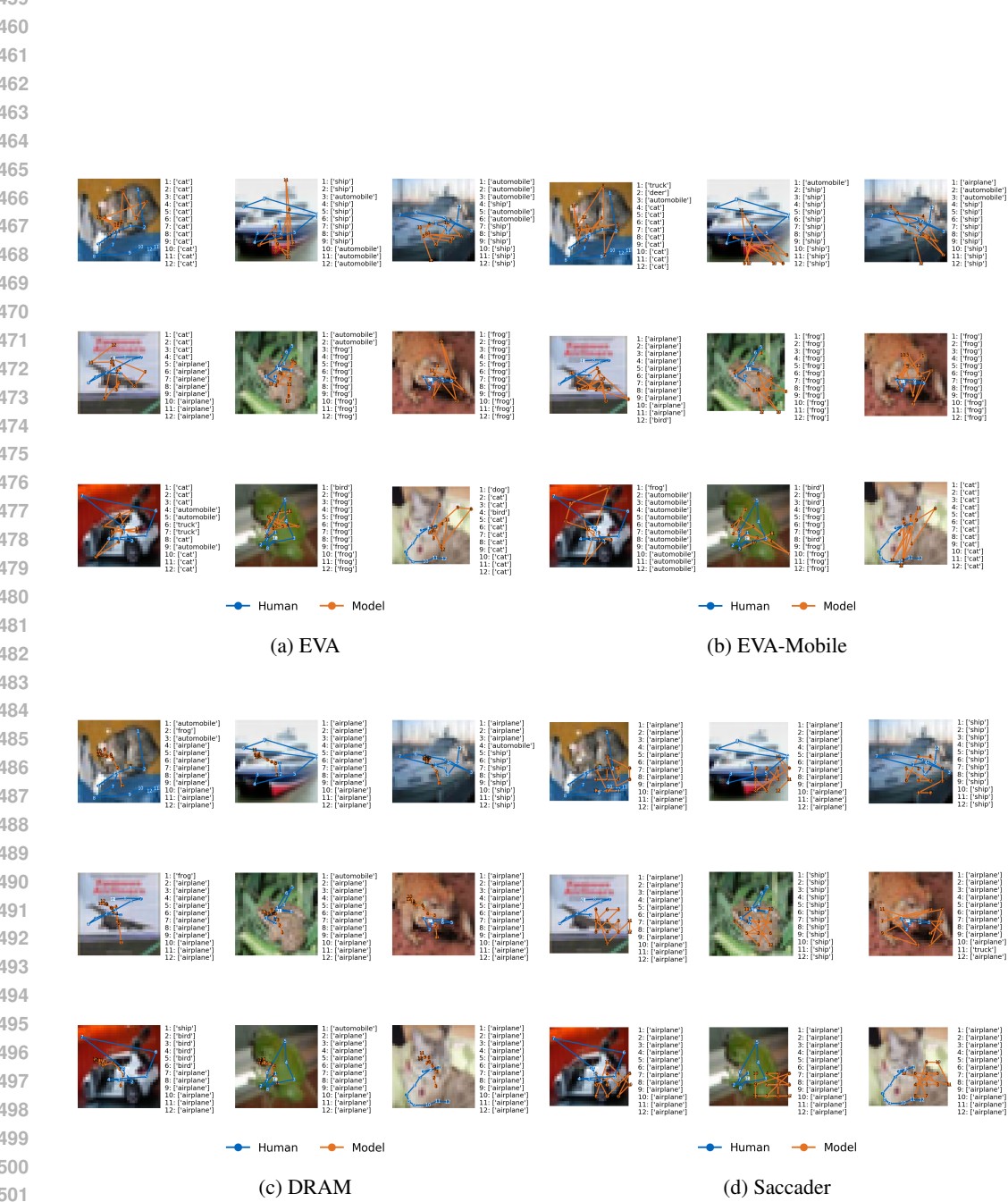

Figure Supp.14: Comparison of scanpath on CIFAR-10 with predictions.

