# OpenReview forum: "EVA: Emergent Human-Like Visual Scanpaths in Hard Attention Models"
_ICLR.cc/2026/Conference — Submitted to ICLR 2026_

### Official Review · Reviewer_wFph · 2025-10-26

**Soundness:** 1
**Presentation:** 2
**Contribution:** 2
**Rating:** 2
**Confidence:** 3

**Summary:**

This paper presents EVA, a brain-inspired hard attention model that can perform image classification while also producing scanpaths that align with human gaze patterns—without explicit gaze supervision. The experiments show improvements against existing hard-attention baselines on classification and scanpath prediction.

**Strengths:**

- The brain-inspired components in the hard attention model is interesting.
- EVA performs better than previous hard attention models on image classification and scanpath prediction.

**Weaknesses:**

- The dataset scale is limited. The experiments are conducted on CIFAR-10 and ImageNet-10, which are relatively small and low-resolution. It is unclear whether EVA can scale to larger datasets, e.g. COCO, SALICON.
- The presentation of the results of the scanpath prediction is confusing, the content in table is inconsistent with the text in the manuscript. There are multiple ambiguities too. These make it difficult to understand the results.
- The baselines for both image classification and scanpath prediction are insufficient. More recent models for image classification and scanpath prediction should be compared on larger datasets.
- The analysis on scanpath prediction is insufficient. Some variants of EVA perform better than EVA, there is no discussion about it.

**Questions:**

- What is the reward function?
- Some ablated EVAs are not clear, what are EVA-Mobile, EVA (CNN only), EVA (gate only), EVA (error only) and EVA (train self-error)?
- EVA does not outperform its ablated versions consistently. For example, EVA w/o CNN consistently performs better than EVA, EVA gate only is comparable with EVA w/o CNN. A more comprehensive analysis is needed to better understand the components in EVA.
- The authors need to explain the gaze policies of center fixed, corner-fixed, and shuffled.
- DeepGaze III does not achieve higher scores as shown in table 2.

---

> ### Author Response · Authors · 2025-11-25
> **Thank you and clarification on scalebility & readability**
>
> We thank the reviewer for carefully reading our paper and for the constructive suggestions. We address the main concerns below and indicate the corresponding changes in the revised manuscript.
>
> **W1** Dataset scale and real-world relevance
>
> We agree that our original presentation did not sufficiently clarify scope and scalability. In the revised paper we explicitly position our core contribution as an interpretability / human-alignment study of hard attention in image classification with gaze, for which CIFAR-10&Gaze-CIFAR-10 is the only setting where both labels and task-aligned scanpaths are available.
>
> Because training and analyzing EVA end-to-end on full ImageNet-1k is difficult within the rebuttal period, we instead added experiments on ImageNet-100 using the EVA-Mobile variant (MobileNetV3 backbone). The results (now reported in the appendix C.) show that EVA maintains a consistent accuracy/compute trade-off relative to CNN and hard-attention baselines, indicating that the architecture does scale to more diverse and higher-resolution images.
>
> As we mention in the discussion, the neuromodulator in EVA is explicitly defined on single-label classification error across glimpses.  Adapting EVA for object detection would require non-trivial redesign of the policy and reward, and we therefore consider it interesting future work rather than part of this paper’s core claim. We now clearly state that large-scale detection/search tasks (COCO-Search18) are beyond the primary scope of this work and are treated only as preliminary evidence in the appendix.
>
> **W2** Confusing scanpath results and table inconsistencies
>
> We appreciate this feedback and have significantly restructured the presentation. We have checked to make sure the consistency and no ambiguities.
>
> **W3** Baselines for classification and scanpath prediction
>
> In this work, we mainly compare between hard attention models where model can generate both scanpath and prediction. To the best of our knowledge, Saccader model is the state-of-the-art hard attention model, but not end-to-end.  Saccader is re-implemented with a ResNet or MobileNet backbone (rather than BAGNet) and evaluated under the same glimpse schedule as EVA for a fair comparison. Additionally, we add soft attention baselines like ViT-tiny and going to add HAT or Gazeformer as SOTA scanpath baseline. We used MobileNetV3 as the latest baseline in both efficient computing and SOTA performance. Details are now given in Sec. 4.1 and the appendix D.
>
> **W4&Q2&Q3** Analysis of scanpath prediction and ablations, ablated EVAs are not clear, EVA does not outperform its ablated versions consistently
>
> We thank the reviewer for pointing this out; we have clarified this trade-off in the revised text.
>
> We now emphasize that our goal is not to maximize SS or accuracy at any cost, but to find a balanced trade-off between classification accuracy and human-alignment. Some ablations (e.g., EVA w/o CNN) produce slightly higher SS but at the price of lower accuracy or unstable training.
>
> In Sec. 4.1 (“Ablation studies”) we now explicitly describe:
> * EVA w/o CNN. Removing the foveal CNN, weakens feature quality and drops accuracy, but can improve SS.
> * EVA gate-only / error-only. Using only the gate or only the neuromodulator yield strong SS but hurts accuracy or destabilizes optimization.
> * EVA-Mobile. Replacing the small CNN with a pretrained MobileNetV3 leads to better SS (stronger features) but slightly worse accuracy than EVA.
> * EVA is chosen as the “main” model because it provides the best overall trade-off, not because it dominates all variants on every individual metric.
>
> **Q1**. What is the reward function?
> EVA is trained with a combination of (i) a standard cross-entropy classification loss and (ii) a REINFORCE term on the glimpse locations. The reward function is now in the revised version we define in Sec. 3.5:
>
> **Q4**. Gaze policies in Table 2 (center-fixed, corner-fixed, shuffled).
> We will clarified these definitions in Appendix in our next revision because we are running out of room:
>
> * Center-fixed: All glimpses are forced to the image center at every time step.
> * Corner-fixed: Glimpses cycle through the four image corners in a fixed pattern, independent of the image content.
> * Random: Each glimpse location is sampled randomly from a uniform distribution over the image.
> * Shuffled: We take EVA’s predicted locations but shuffle their order across time, destroying temporal structure while keeping the same set of points.
> These perturbations probe different aspects of the learned policy (e.g., reliance on central bias vs. specific temporal strategies).
>
> **Q5** Why DeepGaze III does not achieve higher scores.
> DeepGaze III is trained for free-viewing saliency prediction, not task-driven classification. On Gaze-CIFAR-10, it produces almost identical center-biased saliency maps regardless of the target class.
>
> We hope these clarifications address the reviewer’s concerns.
>
> Best regards,
>
> The Authors

---

> ### Author Response · Authors · 2025-11-27
> **Additional results on ImageNet-100 and COCO-Search18 (Same in the revised version of Appendix)**
>
> We thank the reviewers again for their thoughtful feedback. Following the concerns about dataset scale, real-world relevance, we have completed two additional sets of experiments. These do not change the core claims of the paper, but they provide more concrete evidence that EVA and its variants with MobileNet as backbone (i) scale beyond CIFAR-10 and (ii) retain meaningful, human-aligned scanpaths on a more realistic task. We briefly summarise the main numbers below; full details and discussion are included in the new revision (v3) of the manuscript.
>
> **1. ImageNet-100 classification (EVA-Mobile vs. hard-attention baselines)**
>
> On ImageNet-100, EVA-Mobile (with a pretrained MobileNetV3 backbone) achieves strong accuracy while remaining in the same compute regime as other hard-attention models. The table below reports FLOPs per image and Top-1 / Top-5 accuracy:
>
> | Model                 | FLOPs (B) | Top-1 Acc. (%) | Top-5 Acc. (%) |
> |-----------------------|-----------|----------------|----------------|
> | MobileNet (scratch)   | 2.2       | 47.64          | 76.12          |
> | MobileNet (pre.)      | 2.2       | 80.36          | 96.50          |
> | RAM                   | 0.35      | 11.26          | 32.78          |
> | DRAM                  | 84.96     | 21.44          | 47.30          |
> | MRAM                  | 0.35      | 12.88          | 34.76          |
> | **EVA-Mobile (pre.)** | 8.87      | **71.92**      | **91.92**      |
> | EVA                   | 6.60      | 65.86          | 80.24          |
>
> EVA-Mobile (pre.) substantially outperforms prior hard-attention models (RAM/DRAM/MRAM) and narrows the gap to the full-image MobileNet baseline, while still operating with a limited number of glimpses and comparable FLOPs. This supports our claim that the neuromodulated gating architecture scales beyond CIFAR-10 to more diverse, higher-resolution images.
>
> **2. COCO-Search18: preliminary scanpath results**
>
> We also ran a pilot study on the COCO-Search18 dataset (10-class setting), treating it as a goal-directed search and classification task. We report accuracy on standard COCO validation images and on COCO-Search18 (test data is unavailable), together with DTW (↓), ScanMatch (SM ↑), NSS (↑), and AUC (↑) between model scanpaths and human search fixations:
>
> | Model                   | COCO Acc. % ↑ | COCO-search Acc. % ↑ | DTW ↓   | SM ↑   | NSS ↑   | AUC ↑  |
> |-------------------------|---------------|----------------------|---------|--------|---------|--------|
> | CNN MobileNet (pre.)    | 58.82         | 27.50                | –       | –      | –       | –      |
> | Gazeformer    |–         | –                | 168.39      | 0.571      | 1.961       | 0.8      |
> | RAM                     | 34.81         | 12.83                | 500.01  | 0.072  | -0.132  | 0.585  |
> | DRAM                    | 43.32         | 14.34                | 530.78  | 0.077  | -0.124  | 0.587  |
> | MRAM                    | 35.86         | 14.17                | 624.57  | 0.015  | -0.090  | 0.605  |
> | EVA-Mobile (scratch)    | 45.81         | 14.79                | 513.48  | 0.101  | -0.074  | 0.593  |
> | **EVA-Mobile (pre.)**   | 55.82         | 16.63                | **280.29** | **0.313** | **0.307** | **0.714** |
>
> Here EVA-Mobile (pre.) again provides the best overall trade-off between task performance and human-alignment: it achieves the highest COCO and COCO-Search accuracies among **hard-attention models**, substantially lower DTW (better temporal alignment), and much stronger SM/NSS/AUC, that is also competitive to a supervised modern baseline, Gazeformer. While a full object-detection pipeline is beyond the scope of this paper, these results suggest that EVA’s gaze policy remains object-centred and human-aligned even on a more realistic search dataset.
>
> We also provide visulization results on this depository:
> https://anonymous.4open.science/r/Anon-EVA-8607
>
> We hope these additional results help address the remaining concerns about scalability and scanpath evaluation. If time permits, we would greatly appreciate any further feedback on whether these additions clarify the strengths or limitations of EVA.

---

### Official Review · Reviewer_rNCf · 2025-10-31

**Soundness:** 2
**Presentation:** 1
**Contribution:** 2
**Rating:** 4
**Confidence:** 2

**Summary:**

- The draft proposes EVA, a lightweight neuro-inspired hard-attention vision model that classifies images by making a small number of sequential glimpses rather than processing the whole image.
- At each step, it extracts a high-resolution foveal crop plus low-resolution peripheral context, encodes them with a tiny CNN, and updates two cooperating RNNs: a locator that predicts the next fixation and a classifier that integrates evidence.
- A key idea is an uncertainty-modulated saccade policy—the model widens or tightens its fixation sampling variance based on EMA of prediction error—paired with a pulvinar-style gating mechanism that blends top-down state with bottom-up evidence to stabilize exploration and recognition.
- EVA delivers good accuracy and computational efficiency while producing human-like scanpaths (quantified with DTW/ScanMatch/NSS/AUC). It passes stress tests where fixing or shuffling gaze degrades performance, indicating the learned policy is genuinely task-useful.

**Strengths:**

1. Clean ablations that tie each design choice to both accuracy and scanpath alignment.
2. Robustness & qualitative alignment: The model shows stable performance under gaze perturbations (degradation patterns consistent with a learned, task-useful policy) and provides qualitative scanpath visualizations that align with salient object regions; useful for interpretation and practitioner trust.
3. Practical efficiency: Few-glimpse inference yields a solid accuracy/compute trade-off, and the uncertainty-driven exploration + gating are lightweight, easily adoptable ideas for other sequential tasks.

**Weaknesses:**

1. Narrow evaluation scope: Results are confined to CIFAR-10 and a curated ImageNet-10 subset. ImageNet-10 is supposedly used only for evaluation, as it contains 20 images per class. There is no justification for the choice or usage of this dataset. Claims about general human-like scanpaths and efficiency don’t transfer without broader tests.
2. Clarity/consistency gaps: Undefined symbols, figure–text mismatches, and incomplete equations weaken the narrative and justification for the role of every module, like:

A. The figure and equations mismatch

B. $L_{REINFORCE}$ details are missing. The final objective, along with classification loss, is also not described.

C. The significance of foveal and peripheral crops is not explored. There is proper motivation/ role of attention before gating.

D.The composite SS score is built from ad-hoc weights (including a negative weight for DTW), and the paper does not specify how D/S/N/A are normalized. This makes overall rankings fragile to the chosen scaling and weights, so conclusions about “emergent human-like gaze behavior” may be artifacts of aggregation rather than genuine improvements.

3. Lack of reproducibility.

**Questions:**

1. Due to the figure and context mismatch, it's difficult to follow sections 3.1 and 3.2.
Are $x^p_t$ and $x^f_t$ cropped images or vectors?
How is it concatenated if it's a crop, or how is it passed to CNN if it's a vector?
$f_t$, which is the output of the CNN, is nowhere mentioned in the equations. Please explain this in detail with the right flow of events. It becomes challenging to understand how foveal and peripheral vision are justified in the face of this confusion.

2. SS weighting is heuristic and self-negating: with DTW as lower-better, weight −0.25 can cancel out other gains. Prefer to invert DTW first (e.g., (1−DTWnorm)×0.25) and keep all weights ≥0.

3. Clarify normalization for D, S, N, A (DTW, ScanMatch, NSS, AUC), and pre- & post-transform ranges.

4. What is w/o error, error only, and train self-error in the ablations?

5. What is the motivation for Eqs. 11–15? What is the output term finally?

6. How is $L_{REINFORCE}$ loss defined in your case? Also, I assume cross-entropy loss is applied for classification. But exactly on what term is it applied? Also, what is y_t?

Minor corrections

1. Define retina, LGN, f_g, (in figure) and correct r_t → h_t.
2. $f_g$ and $g_t$ in the figure do not match the equations
3. Table 3 is not referred to anywhere in the text.
4. Mention the computational requirements and training details, like hyperparameter settings.
5. Number of random seed experiments conducted and (mean, std) values.
6. Flops calculated are per glimpse or across all glimpses?

---

> ### Author Response · Authors · 2025-11-25
> **Thank you and clarification on architecture, SS metric & reproducibility**
>
> We thank the reviewer for the thoughtful and constructive feedback, and for recognizing both the motivation (emergent scanpaths from task-driven RL) and the principled, biologically grounded structure of EVA.
>
> **W1**. Narrow evaluation scope
>
> In the revised manuscript we explicitly restrict the core claims to image classification with gaze on CIFAR-10 / Gaze-CIFAR-10, where aligned labels & scanpaths actually exist. We now stress that the main contribution is an interpretability study of hard attention, not a generic large-scale benchmark.
>
> Because running full ImageNet-1K + gaze within the rebuttal window is unrealistic, we extended experiments to ImageNet-100 using EVA-Mobile. The results (reported in the appendix) show that EVA retains a favorable accuracy/compute trade-off relative to CNN and hard-attention baselines on a more diverse, higher-resolution dataset. This supports that the architecture scales beyond CIFAR-10 at least on classification.
>
> **W2&Q8**. Clarity/consistency gaps
>
> (A) Figure–equation mismatch
>
> We redesigned Fig. 1–2 so that every symbol in the diagram now matches the notation in Sec. 3.1–3.3.
>
> (C) Foveal vs. peripheral
>
> We fully agree that this was under-explained. This design is to mimic human vision, where we shows different dynamics of 1-scale MRAM and 2-scale MRAM in Figure Supp.7-8, that the foveal and peripheral crops (2-scale) stables the higher-layer RNN dynamic.
>
> **W3**. Lack of reproducibility.
>
> Appendix D describes details of model settings, random seed of the experiments. Upon acceptance, the source code of this work will be released via Github.
>
> **Q1**. figure and context mismatch.
>
> We thank the reviewer for pointing out the confusion in Sec. 3.1–3.2. In the original draft the notation overloaded $g_t$, which made it unclear whether $x^p_t$ and $x^f_t$ were crops or vectors and how they were combined.
> $x^p_t$ and $x^f_t$ are image crops from peripheral and foveal. We modified the Equation 3 and related descriptions to clarify that in our revised version.
>
> **Q2&Q3**. The composite SS, weighting is heuristic and self-negating, Clarify normalization
>
> We have completely cleaned up this part, with details in the section 3.6 and section Appendix D following your suggestions.
>
> **Q4**. What is w/o error, error only, and train self-error in the ablations?
>
> We now explicitly describe that in Sec. 4.1 (“Ablation studies”).
>
> **Q5**. What is the motivation for Eqs. 11–15? What is the output term finally?
>
> Eqs. (11–15) are equations for the pulvinar gate define a QKV-style attention gate We added a short paragraph giving intuition: the gate stabilizes exploration by letting the higher layer “decide” how much to rely on new evidence vs. its current state. The final output is $z_t$, as the input to the higher layer RNN.
>
> **Q6**. How is REINFORCE loss defined in your case?
>
> We fully specify the objective in the training section 3.5.
>
> **Minor corrections**.
> * Define retina, LGN, f_g, (in figure) and correct r_t → h_t.
>
> retina, LGN, f_g are better defined in figure 2. r_t is also corrected as h_t.
>
> * Table 3 is not referred to anywhere in the text.
>
> Table 3 is now relocationed to Appendix, and refered there.
>
> * Mention the computational requirements and training details, like hyperparameter settings.
>
> The details of the parameters, and computational requirements are listed in Appendix D.
>
> * Number of random seed experiments conducted and (mean, std) values.
>
> The random seed used in experiment is described in the Appendix E, we will expand the experiment with multiple random seed and  (mean, std) values and report in our final version.
>
> Computational requirements, training schedule, and seed choices are described in the experimental details Appendix D..
>
> * Flops calculated are per glimpse or across all glimpses?
>
>  Flops calculated are across 12 glimpses for 9 batches in all hard attention models, the details are in Appendix D.
>
>
> We are grateful for your detailed and technically sharp feedback. Many of your comments directly improved the clarity of the architecture description, the robustness of our SS metric, and the reproducibility of the work. We hope the revised version better reflects the strengths you identified: clean ablations, robustness to gaze perturbations, and lightweight neuromodulator/gating mechanisms, while addressing the concerns about clarity, aggregation, and scope.
>
> Best regards,
>
> The Authors

---

> ### Author Response · Authors · 2025-11-27
> **Additional results on ImageNet-100 and COCO-Search18 (Same in the revised version of Appendix)**
>
> We thank the reviewers again for their thoughtful feedback. Following the concerns about dataset scale, real-world relevance, we have completed two additional sets of experiments. These do not change the core claims of the paper, but they provide more concrete evidence that EVA and its variants with MobileNet as backbone (i) scale beyond CIFAR-10 and (ii) retain meaningful, human-aligned scanpaths on a more realistic task. We briefly summarise the main numbers below; full details and discussion are included in the new revision (v3) of the manuscript.
>
> **1. ImageNet-100 classification (EVA-Mobile vs. hard-attention baselines)**
>
> On ImageNet-100, EVA-Mobile (with a pretrained MobileNetV3 backbone) achieves strong accuracy while remaining in the same compute regime as other hard-attention models. The table below reports FLOPs per image and Top-1 / Top-5 accuracy:
>
> | Model                 | FLOPs (B) | Top-1 Acc. (%) | Top-5 Acc. (%) |
> |-----------------------|-----------|----------------|----------------|
> | MobileNet (scratch)   | 2.2       | 47.64          | 76.12          |
> | MobileNet (pre.)      | 2.2       | 80.36          | 96.50          |
> | RAM                   | 0.35      | 11.26          | 32.78          |
> | DRAM                  | 84.96     | 21.44          | 47.30          |
> | MRAM                  | 0.35      | 12.88          | 34.76          |
> | **EVA-Mobile (pre.)** | 8.87      | **71.92**      | **91.92**      |
> | EVA                   | 6.60      | 65.86          | 80.24          |
>
> EVA-Mobile (pre.) substantially outperforms prior hard-attention models (RAM/DRAM/MRAM) and narrows the gap to the full-image MobileNet baseline, while still operating with a limited number of glimpses and comparable FLOPs. The FLOPs of MobileNet is updated because we used MobileNetV3large, and the original value is on MobileNetV3small. This supports our claim that the neuromodulated gating architecture scales beyond CIFAR-10 to more diverse, higher-resolution images.
>
> **2. COCO-Search18: preliminary scanpath results**
>
> We also ran a pilot study on the COCO-Search18 dataset (10-class setting), treating it as a goal-directed search and classification task. We report accuracy on standard COCO validation images and on COCO-Search18 (test data is unavailable), together with DTW (↓), ScanMatch (SM ↑), NSS (↑), and AUC (↑) between model scanpaths and human search fixations:
>
> | Model                   | COCO Acc. % ↑ | COCO-search Acc. % ↑ | DTW ↓   | SM ↑   | NSS ↑   | AUC ↑  |
> |-------------------------|---------------|----------------------|---------|--------|---------|--------|
> | CNN MobileNet (pre.)    | 58.82         | 27.50                | –       | –      | –       | –      |
> | Gazeformer    |–         | –                | 168.39      | 0.571      | 1.961       | 0.8      |
> | RAM                     | 34.81         | 12.83                | 500.01  | 0.072  | -0.132  | 0.585  |
> | DRAM                    | 43.32         | 14.34                | 530.78  | 0.077  | -0.124  | 0.587  |
> | MRAM                    | 35.86         | 14.17                | 624.57  | 0.015  | -0.090  | 0.605  |
> | EVA-Mobile (scratch)    | 45.81         | 14.79                | 513.48  | 0.101  | -0.074  | 0.593  |
> | **EVA-Mobile (pre.)**   | 55.82         | 16.63                | **280.29** | **0.313** | **0.307** | **0.714** |
>
> Here EVA-Mobile (pre.) again provides the best overall trade-off between task performance and human-alignment: it achieves the highest COCO and COCO-Search accuracies among **hard-attention models**, substantially lower DTW (better temporal alignment), and much stronger SM/NSS/AUC, that is also competitive to a supervised modern baseline, Gazeformer. While a full object-detection pipeline is beyond the scope of this paper, these results suggest that EVA’s gaze policy remains object-centred and human-aligned even on a more realistic search dataset.
>
> We also provide visulization results on this depository:
> https://anonymous.4open.science/r/Anon-EVA-8607
>
> We hope these additional results help address the remaining concerns about scalability and scanpath evaluation. If time permits, we would greatly appreciate any further feedback on whether these additions clarify the strengths or limitations of EVA.

---

### Official Review · Reviewer_J29t · 2025-10-31

**Soundness:** 2
**Presentation:** 1
**Contribution:** 2
**Rating:** 4
**Confidence:** 3

**Summary:**

EVA (Emergent Visual Attention) presents a **hard-attention model** meant to reproduce human-like visual scanpaths. It extends classical recurrent-attention frameworks (RAM, MRAM, Saccader) by adding (1) a foveated CNN backbone, (2) a neuromodulator that adapts fixation variance based on uncertainty, and (3) a pulvinar-style gating unit between recurrent layers. Training uses REINFORCE with only class labels. Results are shown on CIFAR-10, ImageNet-10, and a small gaze dataset (Gaze-CIFAR-10). The authors report improved accuracy/efficiency over older hard-attention baselines and higher scanpath similarity to human gaze trajectories.

**Strengths:**

1.  The goal of demonstrating that human-like scanpaths can emerge naturally from task-driven reinforcement learning is well motivated and connects computational attention models with established ideas from active vision and neuroscience.

2.  The incorporation of neuromodulator and pulvinar-inspired modules provides a biologically grounded mechanism for adaptive uncertainty control and selective information routing, giving the overall model a principled structure.

**Weaknesses:**

-   The dataset choice limits the paper’s real-world relevance. Evaluations on small or reduced subsets (e.g., CIFAR-10, ImageNet-10) do not fully reflect the diversity and structural richness of natural scenes in ImageNet-1K, where distinct foreground–background composition and color variation would make scanpath behavior more meaningful and comparable to human gaze.

-   Figure 3 is difficult to interpret. The visual layout and captioning are unclear. Figure 1 is also confusing; the left portion is not well explained, and the caption does not clearly describe what each component represents. Captions throughout the paper are inconsistently written, reducing overall readability.

-   The model design relies on CNN-RNN-style sequencing rather than transformer-based attention architectures. Since transformers inherently support sequential and global context modeling, a comparison or discussion relative to such architectures is needed to position the work within modern attention frameworks.

- The experimental scope is narrow. Results are limited to classification datasets, and the paper does not test EVA on tasks where scanpath modeling or attention dynamics would be most relevant (e.g., visual question answering, referring expressions, or object search).

**Questions:**

- It is unclear how the Saccader baseline was adapted for the reported experiments. The original Saccader model was designed and trained on high-resolution ImageNet images (224×224 and above) with multi-saccade inference. The paper evaluates on smaller datasets such as CIFAR-10 and ImageNet-10, which differ significantly in scale and visual statistics. Details about how Saccader was reconfigured are missing. Without this clarification, the fairness of the comparison remains uncertain.

- Recent works such as Zero-TPrune (CVPR 2024) [1] and MADTP (CVPR 2024) [2] demonstrate adaptive token selection in vision transformers, achieving spatial efficiency similar in spirit to EVA’s selective attention. It would be important for the authors to clarify how EVA compares to these more recent adaptive-computation baselines, which now represent the standard for efficient attention modeling.

- Please address the points mentioned in the Weaknesses section as well.

[1]. Zero-TPrune: Zero-Shot Token Pruning through Leveraging of the Attention Graph in Pre-Trained Transformers, CVPR 2024
[2]. MADTP: Multimodal Alignment-Guided Dynamic Token Pruning for Accelerating Vision-Language Transformer, CVPR 2024

---

> ### Author Response · Authors · 2025-11-25
> **Thank you and clarifications on scalability, datasets, and interpretability**
>
> We thank the reviewer for the thoughtful and constructive feedback, and for recognizing both the motivation (emergent scanpaths from task-driven RL) and the principled, biologically grounded structure of EVA.
>
> **W1&W4**. Dataset scale and real-world relevance
>
> We agree that CIFAR-10 and ImageNet-10 are small compared to ImageNet-1K and COCO. Our goal in this paper, however, is not to compete with large-scale recognition systems, but to answer a more focused interpretability question: **can a hard-attention model trained only on class labels learn a gaze policy that (i) is task-useful and (ii) aligns with human scanpaths?**
>
> We therefore chose Gaze-CIFAR-10, which is (to our knowledge) the only dataset that jointly provides image-classification labels and sequential human scanpaths on the same task, enabling a controlled comparison between model and human gaze.
>
> In the revised manuscript, we explicitly frame CIFAR-10 as the primary benchmark for alignment and interpretability, and move ImageNet and COCO-Search18 to the appendix as scalability checks rather than main contributions.
>
> Due to computational constraints, we were not able to complete experiments on full ImageNet-1K within the rebuttal period. Instead, we ran additional experiments on ImageNet-100, where EVA also achieves strong classification accuracy and maintains meaningful, object-centred scanpaths (see App. C). In contrast, adapting EVA to full object-detection pipelines (e.g., COCO) would require substantial architectural changes (e.g., multi-object outputs, task-specific reward design), which we consider beyond the scope of this first study focused on image classification with human gaze comparison.
>
> More diverse tasks (VQA, referring expressions, search) are important future work, but we hope this clarifies that the current scope is narrow but intense: a first controlled study of emergent, human-like scanpaths in hard attention, rather than a general-purpose large-scale detector.
>
> **W2**. Figure clarity and captions (former Fig. 1 / Fig. 3)
>
> We appreciate the comments on figure readability and have substantially revised the visuals:
>
> * Former Fig. 1 (concept) has been redrawn with clearer presentation on the concept of this work.
> * Former Fig. 3 (scanpaths) has been replaced by a new figure that shows human vs model scanpaths side-by-side for several examples and is aligned in a row/column layout. The caption explicitly defines colours and what each panel represents.
>
> We also went through all captions to make terminology and style consistent, and ensured that every symbol appearing in the figures is defined in the main text.
>
> We hope this addresses the concerns about visual layout and captioning.
>
> **W3&Q2**. Positioning relative to transformers and adaptive token pruning
>
> We agree that modern attention work is heavily transformer-based, we add transformer-based models as baselines, and we will add clarification in the related-work section in next revision:
> Conceptually, the visual attention models in mordern computer visions can be divided to soft attention and hard attention [1], and our works are in the stochastic, reinforce-based hard attention models. In the meanwhile, transformer models are one of the soft attention. The key difference between our model to transformer is that transformer takes every pixel of the image for computation, while our model only processes limited pixels.
>
> As for the difference between the recent work on transformer-based models: Zero-TPrune and MADTP. Our works actually shares similar ideas, yet our core is different. We think models like Zero-TPrune are top-down models, that pruning the transformer for efficient computation on the full image. Our method are more like a bottom-up fashion, that EVA makes continuous, spatial fixation choices before high-resolution features are computed, operating directly in pixel space through a human-like foveal-peripheral mechanism. This means EVA never observes the whole high-res image at once, while token-pruning ViTs do.
>
> As a result, EVA and token-pruning transformers are complementary: EVA focuses on biologically inspired, sequential sampling and human-aligned scanpaths while adaptive token pruning focuses on efficiency within pre-trained global transformers. Combining the two (e.g., using a neuromodulated policy to control token pruning) is an interesting direction for future work, but beyond the current paper’s scope.
>
> **Q1**. It is unclear how the Saccader baseline was adapted for the reported experiments.
>
> We apologize for the lack of detail in the original submission and have added a dedicated paragraph in App. D describing our Saccader configuration.
>
> We hope these clarifications and newly added discussions make our positioning within the broader attention literature clearer.
>
> [1] Hassanin et al., Visual Attention Methods in Deep Learning: An In-Depth Survey, arXiv, 2024
>
> Best regards,
>
> The Authors

---

> ### Author Response · Authors · 2025-11-27
> **Additional results on ImageNet-100 and COCO-Search18 (Same in the revised version of Appendix)**
>
> We thank the reviewers again for their thoughtful feedback. Following the concerns about dataset scale, real-world relevance, we have completed two additional sets of experiments. These do not change the core claims of the paper, but they provide more concrete evidence that EVA and its variants with MobileNet as backbone (i) scale beyond CIFAR-10 and (ii) retain meaningful, human-aligned scanpaths on a more realistic task. We briefly summarise the main numbers below; full details and discussion are included in the new revision (v3) of the manuscript.
>
> **1. ImageNet-100 classification (EVA-Mobile vs. hard-attention baselines)**
>
> On ImageNet-100, EVA-Mobile (with a pretrained MobileNetV3Large backbone) achieves strong accuracy while remaining in the same compute regime as other hard-attention models. The table below reports FLOPs per image and Top-1 / Top-5 accuracy:
>
> | Model                 | FLOPs (B) | Top-1 Acc. (%) | Top-5 Acc. (%) |
> |-----------------------|-----------|----------------|----------------|
> | MobileNet (scratch)   | 2.2       | 47.64          | 76.12          |
> | MobileNet (pre.)      | 2.2       | 80.36          | 96.50          |
> | RAM                   | 0.35      | 11.26          | 32.78          |
> | DRAM                  | 84.96     | 21.44          | 47.30          |
> | MRAM                  | 0.35      | 12.88          | 34.76          |
> | **EVA-Mobile (pre.)** | 8.87      | **71.92**      | **91.92**      |
> | EVA                   | 6.60      | 65.86          | 80.24          |
>
> EVA-Mobile (pre.) substantially outperforms prior hard-attention models (RAM/DRAM/MRAM) and narrows the gap to the full-image MobileNet baseline, while still operating with a limited number of glimpses and comparable FLOPs. The FLOPs of MobileNet is updated because we used MobileNetV3large, and the original value is on MobileNetV3small. This supports our claim that the neuromodulated gating architecture scales beyond CIFAR-10 to more diverse, higher-resolution images.
>
> **2. COCO-Search18: preliminary scanpath results**
>
> We also ran a pilot study on the COCO-Search18 dataset (10-class setting), treating it as a goal-directed search and classification task. We report accuracy on standard COCO validation images and on COCO-Search18 (test data is unavailable), together with DTW (↓), ScanMatch (SM ↑), NSS (↑), and AUC (↑) between model scanpaths and human search fixations:
>
> | Model                   | COCO Acc. % ↑ | COCO-search Acc. % ↑ | DTW ↓   | SM ↑   | NSS ↑   | AUC ↑  |
> |-------------------------|---------------|----------------------|---------|--------|---------|--------|
> | CNN MobileNet (pre.)    | 58.82         | 27.50                | –       | –      | –       | –      |
> | Gazeformer    |–         | –                | 168.39      | 0.571      | 1.961       | 0.8      |
> | RAM                     | 34.81         | 12.83                | 500.01  | 0.072  | -0.132  | 0.585  |
> | DRAM                    | 43.32         | 14.34                | 530.78  | 0.077  | -0.124  | 0.587  |
> | MRAM                    | 35.86         | 14.17                | 624.57  | 0.015  | -0.090  | 0.605  |
> | EVA-Mobile (scratch)    | 45.81         | 14.79                | 513.48  | 0.101  | -0.074  | 0.593  |
> | **EVA-Mobile (pre.)**   | 55.82         | 16.63                | **280.29** | **0.313** | **0.307** | **0.714** |
>
> Here EVA-Mobile (pre.) again provides the best overall trade-off between task performance and human-alignment: it achieves the highest COCO and COCO-Search accuracies among **hard-attention models**, substantially lower DTW (better temporal alignment), and much stronger SM/NSS/AUC, that is also competitive to a supervised modern baseline, Gazeformer. While a full object-detection pipeline is beyond the scope of this paper, these results suggest that EVA’s gaze policy remains object-centred and human-aligned even on a more realistic search dataset.
>
> We also provide visulization results on this depository:
> https://anonymous.4open.science/r/Anon-EVA-8607
>
> We hope these additional results help address the remaining concerns about scalability and scanpath evaluation. If time permits, we would greatly appreciate any further feedback on whether these additions clarify the strengths or limitations of EVA.

---

### Official Review · Reviewer_1qs9 · 2025-11-01

**Soundness:** 3
**Presentation:** 2
**Contribution:** 3
**Rating:** 6
**Confidence:** 4

**Summary:**

The authors introduce EVA, a hard attention model that is designed with the human brain as inspiration. Trained with reinforcement learning on class labels, EVA proves to be competitive in object recognition task while being efficient and also aligns well with human gaze even without training explicitly on eye gaze data. These emergent gaze behavior within EVA is claimed by authors to be a significant step towards model interpretability.

**Strengths:**

1. Appreciable model design: The model design is technically sound and biologically plausible. EVA’s model design can be a good recipe for other brain-inspired vision models. The authors show that only training EVA on class labels using reinforcement learning can induce emergent gaze behavior, which does serve testament to EVA being biologically plausible, and therefore can further be probed to investigate several cognitive tasks.
2. SOTA performance: EVA achieves a reasonable tradeoff between performance and efficiency across several variants.
3. Exhaustive experiments: The ablations and experiments are adequate and align with the claims of the paper.

**Weaknesses:**

1. Weak motivation: I do not agree with the central motivation for this work that is mentioned in L 49-51  - “We argue that for AI systems to become interpretable and reliable partners, they must not only perform tasks well and share human-like patterns of attention, they also need to have similar structure with human”. Human scan paths are often not interpretable due to noise and inter-observer variability. I am not convinced if exhibiting gaze patterns improves interpretability of EVA.

2. Claim of interpretability not supported adequately: The claim of “interpretability” has not been explicitly shown in the paper beyond alignment metrics with human gaze behavior and some qualitative examples (which do not showcase interpretability). How do I, as an AI practitioner, interpret these EVA-generated gaze patterns?

3. Poor readability: The overall content of the paper is somewhat disorganized with some elements begging further elaboration, for instance: (a) In lines 309-320, the authors talk about how “EVA achieves the strongest human alignment in human dynamic scanpath” on the basis of the scanpath metrics in Table 2, but introduce these metrics way later in Sec. 4.2. I suggest moving the material around for easier reading. (b) What is the difference between EVA and EVA-Mobile? I am guessing it is the difference in backbone networks, but this has not been explicitly specified in the text. (c) Additionally, there are missing references in L. 724-725.

**Questions:**

1. I have no problem believing DeepGaze-III will not work well on this task, as it is designed for free-viewing behavior, not object recognition. However, I wanted to ask the authors if they considered any other scanpath prediction model to benchmark.

2. Can the authors shed light on the different trends shown in EVA with and without CNN, gate, error etc? Can they also address why EVA-mobile is better aligned with human behavior than EVA?

3. Why is DTW and SM missing for DeepGaze-III in Table 2?

4. I found it hard to parse Table 2 with only EVA's metrics in bold. It is hard to understand and compare varying metrics of baselines and EVA variants. Why is EVA highlighted in bold even though EVA-mobile is almost 50% better in terms of the composite scanpath similarity metric SS?

---

> ### Author Response · Authors · 2025-11-25
> **Thank you and clarification on motivation & interpretability**
>
> We thank the reviewer for the constructive and detailed feedback, and for highlighting the strengths in model design, experimental thoroughness, and biological plausibility. Below we address each concern and describe the corresponding changes in the revised manuscript.
>
> **W1**.“Weak motivation: I do not agree with the central motivation for this work in L 49-51
>
> We agree that human gaze is noisy and exhibits substantial inter-observer variability, and we appreciate the reviewer’s clarification. At the same time, as we note in L59 of the revised paper, prior work has shown that, conditioned on a task, human scanpaths still exhibit consistent structure and task-specific regularities across observers. Our work builds on this line of this evidence.
>
> In the revision we have softened and refined our motivation:
> We removed the strong claim that interpretable AI systems “must … have similar structure with humans.” We now state more cautiously that sharing human-like attention patterns and using human-inspired modules provides one interpretable channel for understanding model behaviour, rather than a necessary condition for interpretability.
>
> **W2**.Claim of interpretability not supported adequately
>
> We want to stress that interpretability in EVA is not solely rely on “looking human-like.” In the our revised Discussion we explicitly frame our evidence as the combination of
> (i) quantitative gaze-alignment metrics,
> (ii) gaze-perturbation experiments (Table 2) showing that performance is tightly coupled to the informative area and learned scanpath, and
> (iii) PCA of hidden states (Fig. 5) revealing structured, class-specific internal dynamics.
>
> Together, these analyses support our claim that EVA offers a behaviorally interpretable attention policy whose sampling strategy can be related to, but not equal to, the human gaze.
>
> **W3**.Poor readability “Content somewhat disorganized… EVA vs EVA-Mobile not specified… missing references.”
>
> We have carefully revised the manuscript to address these points:
> * Reordered scanpath-metric description.
> The composite scanpath similarity metric and its components (DTW, ScanMatch, NSS, AUC) are now part of the Methods (Sec. 3.6), before any results are discussed.
>
> * Clarified EVA vs EVA-Mobile.
> We now clearly define EVA-Mobile in Sec. 4.1. Your sense is correct:
> EVA uses our lightweight foveal CNN described in Figure Supp.2.
> EVA-Mobile replaces this CNN with a MobileNetV3 backbone pretrained on ImageNet, while keeping the neuromodulator and gate unchanged.
> We describe EVA-Mobile as a scaled-up variant in ablation mainly used to test whether stronger foveal features improve scanpath alignment.
>
> * Fixed missing references.
> All missing citations around the previous L.724–725 have been corrected in the revision.
>
> **Q1**.I have no problem believing DeepGaze-III ... I wanted to ask the authors if they considered any other scanpath prediction model to benchmark.
>
> * Yes. We agree that comparing against more recent scanpath models is valuable. We investigated HAT and Gazeformer (CVPR 2023/2024) as potential baselines. However, we encountered several technical issues as we listed in general response. We will uses one of these SOTA models as our scanpath model baseline in our future revision.
>
> **Q2**. Can the authors shed light on ...? Can they also address ...?
>
> * From our PCA visualizations, EVA is different from MRAM mainly in the dynamics of the lower layer RNN. EVA's design in prediction-error modulation and gate mechanisms introduces dynamics to the lower layer RNN for exploration and exploitation. We believe this dynamic in scanpath or gaze movement is what makes the predicted scanpath more closer to human. From the Figure Supp.9, we shows more chaotics in the the lower layer RNN dynamic, and we are positive that this dynamic brings EVA-mobile more human-aligned score than the EVA model.
>
> **Q3**. Why is DTW and SM missing for DeepGaze-III in Table 2?
>
> * From our experiment, we obtained the identical prediction of the fixation points, and we believe this is because of the low resolution of CIFAR image. Therefore, we are inable to calculate these scanpath related metrics (DTW and SM), but only provide saliency based metrics.
>
> **Q4**. I found it hard to parse Table 2 with only EVA's metrics in bold.
>
> * We agree that the original table formatting was confusing. We have revised the table with:
>  shade the rows corresponding to EVA and EVA-Mobile to mark them as our two main models;
> boldface the best value per column across all hard-attention models.
> We hope this makes the comparison much easier to read.
>
> We thank the reviewer again for the insightful comments. We believe our revised manuscript now has clearer motivation, stronger interpretability evidence, improved organization, and more thorough experimental analysis, while preserving the core contribution of EVA as a brain-inspired hard-attention model that jointly advances performance and human-aligned gaze behavior.
>
> Best regards,
>
> The Authors

---

> > ### Comment · Reviewer_1qs9 · 2025-11-27
> > **Thanks for the clarification**
> >
> > Thanks for the clarification, I still have some queries:
> >
> > 1. W1 and W2: I appreciate the change in motivation, however the current motivation is not strong either. However, I am unable to see how scanpaths are more interpretable than say, saliency maps? I believe that it is hard to follow complex scanpaths in Fig. 3. While I appreciate the quantitative gaze-alignment metrics, how do I interpret the model predictions from the emergent scanpaths as an AI practitioner? I am still unable to interpret model predictions through noisy scanpaths.
> >
> > 2. Q1: Can the authors use any other scanpath prediction models like Gazexplain [1] or ScanDiff [2] that might have public implementations?
> >
> > 3. Q3: By "identical prediction", do the authors mean perfect prediction, i.e., did DeepGaze-III predict scanpaths with 100% accuracy?
> >
> >
> > [1] Chen, Xianyu, Ming Jiang, and Qi Zhao. "Gazexplain: Learning to predict natural language explanations of visual scanpaths." European Conference on Computer Vision. Cham: Springer Nature Switzerland, 2024.
> >
> > [2] Cartella, Giuseppe, et al. "Modeling Human Gaze Behavior with Diffusion Models for Unified Scanpath Prediction." Proceedings of the IEEE/CVF International Conference on Computer Vision. 2025.

---

> ### Author Response · Authors · 2025-11-28
> **Response to follow-up on interpretability (W1/W2) and baselines (Q1/Q3)**
>
> We thank the reviewer for the further questions and for pointing us to additional scanpath work. We further address W1/W2 and Q1, Q3 below.
>
> 1. W1 and W2 Scanpaths vs saliency maps; how are they interpretable?
>
> We appreciate the clarification that our motivation still felt too strong. We fully agree that raw scanpaths can be noisy and that long trajectories in Fig. 3 are not always easy to “read by eye” and that this by itself does not make a model more interpretable than standard saliency. Our claim is not that every individual sequence is directly human-readable, but that EVA provides an attention policy whose behaviour can be analysed in a structured way, complementary to saliency maps. We view the scanpath of hard attention models, outgrowth of the classification, particularly useful to analyse exploration strategies and failure modes during classification task, where we show by the following cases:
> * We present the comparison between the model's scanpath and saliency map computed by cv2 and their prediction in Figure Supp.11 and Figure Supp.12. We further present our visualization results on this anonymous depository: https://anonymous.4open.science/r/Anon-EVA-8607
> * We will add detailed descriptions in our revised paper later, that from the saliency map, we see multiple regions that are high salience, but by which order human or an AI model will view is missing from a saliency map alone. Our hard attention model is making classification decisions on every glimpsed croped small image. In our depository, we ploted the visuable region of our model by time, where we can see, on the CIFAR10_foveal_visible_pred_EVA_2.png example, a clear reasonable case at the 1st glimpse to misjudge the the plane to a ship, and at which glimpse, the decisive evidence is forcing the model to change the decision from 1 class to another.
> For an AI practitioner, these sequences help diagnose why the classification model failed in a specific moment: e.g., failures where EVA never fixates the object and failures where it fixates the object but then over-explores distractors. Saliency maps, by contrast, show the union of attended regions but not the temporal strategy.
> However, beyond the scope of the current work, we see an interesting and promising future direction to jointly use saliency maps and scanpaths, leveraging saliency for spatial importance and scanpaths for temporal exploration, to provide richer explanations than either alone.
>
> 2. Q1 – Other scanpath models such as Gazexplain and ScanDiff
>
> Thank you for pointing us to Gazexplain and ScanDiff. Both are highly relevant and we will cite and discuss them explicitly in the related-work section and use one of them as baselines. We already successfully added the Gazeformer as one the baselines, as listed in the following table, and it indeed served as a powerful modern baselines on scanpath prediction, that shows the gap between our emerged scanpath and the scanpath prediction models, positioned us in the modern scanpath prediction models, and we will continue add these models as modern baselines.
>
> | Model                   | COCO Acc. % ↑ | COCO-search Acc. % ↑ | DTW ↓   | SM ↑   | NSS ↑   | AUC ↑  |
> |-------------------------|---------------|----------------------|---------|--------|---------|--------|
> | CNN MobileNet (pre.)    | 58.82         | 27.50                | –       | –      | –       | –      |
> | Gazeformer    |–         | –                | 168.39      | 0.571      | 1.961       | 0.8      |
> | EVA-Mobile (pre.)   | 55.82         | 16.63                | 280.29 | 0.313 | 0.307 | 0.714 |
>
> 3. Q3 – Meaning of “identical prediction” for DeepGaze-III
>
> We apologize for the ambiguity in our wording. We did not mean that DeepGaze-III achieved perfect scanpath prediction (100% aligned). What we observed empirically on Gaze-CIFAR-10 is that, due to the low resolution and strong center bias, DeepGaze-III produced almost the same saliency map and fixation pattern across most images, largely independent of the glimpse steps (no dynamic between steps). We will clarify this in the text by replacing “identical prediction” with a more precise explanation of the observed near-constant center-biased behavior and it prevents meaningful computation of DTW/ScanMatch in our setting.
>
> We are grateful for these follow-up questions, which helped us refine both our claims and our presentation. We hope the revisions and clarified scope will make our interpretability claim more concrete and better grounded.
>
> Best regards,
>
> The Authors

---

### Author Response · Authors · 2025-11-20
**Revision of Manuscript**

We sincerely thank all reviewers for their thoughtful and detailed feedback on our submission.

While we wish to respond in detail to all points raised by each reviewer in detail as soon as possible, we believe it is important to promptly share our key updates: specifically, improvements to our presentation to clarify potential misunderstandings in reviewers, and additional decisive evidence on interpretability, since the central contribution of this paper lies in the interpretability and explainable AI and we are in this right track.

In response to the major concerns raised, we have uploaded Version 2 of the manuscript (PDF) with the following key updates:

**1. Focus and Framing Refined for Clarity**
- In light of reviewer comments we recognised that our original emphasis on achieving state-of-the-art accuracy created confusion about our central contribution, we have revised the title and abstract to emphasise our core contribution: **interpretability and human‐alignment of the attention mechanism, brought by brain-inspired mechanisms, while retaining strong classification performance**.
- We have limited the core evaluation experiments to the image-classification task (CIFAR-10), where human gaze data is available and moved the ImageNet-10 and object‐detection (COCO) experiments to the appendix, explicitly as a scalability check, not the primary contribution.

We have cleaned up some presentations:
- replacing the figure 1 and 2 with detailed explaination and matched mathmatic sign.
- defined all symbols in the training section, including the supervised classification loss and L_REINFORCE (we apologized for the confusion that we didn't wrote this part before because we thought they are not our contribution, and same with previous work, MRAM)
- clarified ablation naming (“CNN only”, “Gate only”, “Error only”) in L. 321-323, L. 362-365
- moved the the section 4.4 COMPOSITE SCANPATH SIMILARITY METRIC to section 3.6 as part of the method.
- rephrased the conclusion and discussion part to keep the paper in 10-page limit.

**2. Interpretability New Evidence**
- We have added a new subsection L. 422- presenting PCA visualisations of the hidden recurrent‐layer states (both lower and higher levels) showing class‐specific trajectories over time. This provides direct interpretability evidence, aligning with our hypothesis about human‐aligned structural dynamics.

**3. Doings**
- We are reconsidering the baseline models for the human scanpath prediction, originally as DeepGaze III model is for free-viewing, and only producing the same fixation location in our experiment. The baselines experiment are ongoing using latest transformers scanpath models(i.e. HAT[1] or Gazeformer), but we are running into some problems like the provided pretrained weights can't be used directly so we have to find a way to train it.
- The cleaning of appendix is still to be completed. We added the visualization of the PCA results of baseline models to Appendix, while they are actually interesting in interpreting the hard attention models. At present, we have completed an experiment setting in COCO-Search dataset. The experiments and results in the Appendix serve primarily as supporting material because the human gaze data on COCO (COCO-search18 and SALICON) are object detection task, that is out of the scope of this work.

Thank you again for your time and constructive suggestions.

[1]. Unifying Top-down and Bottom-up Scanpath Prediction Using Transformers, CVPR 2024 [2]. Gazeformer: Scalable, Effective and Fast Prediction of Goal-Directed Human Attention, CVPR 2023

Best regards,

The Authors

---

> ### Author Response · Authors · 2025-11-25
>
> We sincerely thank all reviewers for their thoughtful and detailed feedback on our submission.
> We have uploaded Version 3 of our manuscript with significant revision to improve the clarity, presentation quality, and results from our additional experiments.
>
> While some of the larger-scale follow-up experiments (e.g., additional tranformer-based scanpath-prediction baselines) are still ongoing and will be reported in a future version, the current revision already incorporates substantial changes to the presentation and adds the key results needed to support our main claims. In what follows, we respond to each reviewer’s comments point-by-point in detail.
>
> Thank you again for your time and constructive suggestions.
>
> Best regards,
>
> The Authors

---

### Author Response · Authors · 2025-12-03
**General response to the Area Chair**

Dear Area Chair,

We would like to sincerely thank you and the reviewers for the time and effort spent evaluating our submission.

Due to the exceptional circumstances of this year’s reviewing process, we unfortunately had very limited opportunity for direct back-and-forth discussion with the original reviewers. Nevertheless, we treated their initial comments seriously and used the rebuttal period to carry out substantial additional work: refining the positioning of the paper, clarifying the methodology, running extra analyses and baselines where feasible, and carefully improving the presentation.

We respectfully ask that you take these efforts into account when forming your judgment. We believe that the revised manuscript now addresses the key questions and weaknesses raised in the reviews and that it offers a substantially clearer, stronger, and more informative contribution than the original submission.

In this note, we briefly summarize how we have addressed the main concerns raised in the reviews. All new or substantially revised parts of the manuscript are highlighted in red so that the changes are easy to inspect.

**Summary of rebuttal**

In response to the reviewers’ comments, we have carefully revised the manuscript along the following directions:

**1. Clarifying motivation, novelty, and positioning.**

*A response to Reviewer 1qs9, that we had overclaiming in some statements in introduction, and most reviewers' complaint on the presentation quality including figure mismatch, ablation naming, section layout, etc.*

We have rewritten parts of the introduction and related work to more clearly articulate the problem we tackle, how our approach differs from existing saliency and scanpath-prediction methods, and how it complements recent human-gaze modeling and hard-attention literature. This addresses concerns about overlap with prior work and clarifies our main contributions. We also add explainations to ablation names, and stability test details.

**2. Methodological clarification, metrics, and technical detail.**

*A response to Reviewer rNCf and wFph, that we had some unclear methodology explainations like what is the reward, retina, LGN etc.*

Several reviewers asked for a clearer explanation of the model components and training pipeline (e.g., how gaze supervision is used, how scanpaths are represented, and how attention trajectories are generated and evaluated). We have significantly expanded the method section, redrawing figures that are more readable and technical correct, and added clarifying text and equations, so that the forward pass, loss functions in training, and evaluation protocol, metrics we used are now easier to follow and reproduce. As reviewer rNCf suggested, we updated our scanpath metrics with new normalization using a upper bound and lower bound whose details are described in Appendix D. now. We updated the result of SS in Table 1 as a consequence of this action.

**3. Experimental coverage and baselines.**

*A response to all Reviewers that CIFAR-10 and ImageNet-10 is not convencing, the interpretability is not fully supported, and the DeepGazeIII baselines is not good.*

Each model has its limitations, and ours is that task is limited to image classification rather than object detection because the model's assumption: every localized glimpse correspond to the same object. Hard attention is still hard to train. Non-trival modification is required to extend this work to other tasks, but we are confident on the feasibility.

During the rebuttal, we have strengthened the experiment section by **(i) adding or updating baselines  for human-gaze alignment (DeepGazeIII for freeviewing to DeepGazeIIE for object recognition, and adding Gazeformer)**, **(ii) reporting results on the ImageNet-100, and COCO-search18 in Appendix C**, and **(iii) providing PCA analysis on the hard attention models**. This directly responds to concerns about the strength, and fairness of baselines, interpretability claim, and helps situate our method with respect to modern gaze/scanpath models.

At last, while addressing reviewers' common concerns and questions, we wish to briefly highlight our main contributions in this work:

1. A lightweight, brain-inspired hard-attention model trained with supervised labels+RL that bridges strong performance and human alignment under limited resource in image classification task.
2. Interpretability: scanpath metrics, the stability test, the robust evaluation, and visualization on the PCAs, and Figure Supp.12 show the model bases decisions aligned with human salient regions.
3. **Emergent human-like gaze & resources**:A scientific finding that EVA exhibits emergent, human-like scanpaths without any gaze supervision, as quantified by multiple metrics. We will release all code and evaluation scripts to facilitate future work in cognitive science and AI on mechanisms of human vision.

Thank you again for your time and consideration.

Sincerely,

The Authors

---

### Meta-Review · Area_Chair_vev4 · 2025-12-23

**Summary:**

The paper introduces EVA, a brain-inspired hard-attention vision model that achieves strong image classification while producing human-aligned gaze patterns and interpretable internal dynamics via sequential foveated glimpses and biologically motivated control mechanisms.

Reviewers raised concerns about overstated claims and presentation quality, as well as insufficient methodological clarity regarding reward design, retinal/LGN modeling, and the overall architecture and training pipeline. Additional concerns focused on limited experimental coverage and baselines, with CIFAR-10 and ImageNet-10 deemed unconvincing, interpretability claims under-supported, and the DeepGazeIII baseline inadequate.

During rebuttal, the authors successfully addressed issues related to writing clarity, claim calibration, and partially strengthened the experimental section by adding updated gaze-alignment baselines, reporting results on ImageNet-100 and COCO-Search18, and providing PCA analyses of attention dynamics. However, the experimental scope remains limited in dataset scale, task diversity, and stress testing across realistic settings, making it difficult to justify the model’s generality or practical deployment.

Moreover, despite claims of biological inspiration, the proposed glimpse mechanism remains a coarse approximation of foveated vision in biological systems, relying on separate central and peripheral streams rather than eccentricity-dependent sampling. Prior work (e.g., [a]) has modeled foveated vision in a more human-like manner. In addition, beyond the baselines suggested by the reviewers, the authors should demonstrate deeper engagement with the literature by including a broader set of relevant models (e.g., [b]). These models could serve both as stronger baselines and as sources of architectural components that may be integrated to make the proposed approach more human-aligned. Overall, the AC concurs with the reviewers that the experimental evidence is insufficient for acceptance at a venue such as ICLR.

References:
[a] Mnih, Volodymyr, et al. "Recurrent models of visual attention." Advances in neural information processing systems 27 (2014).
[b] Wang, Bo, et al. "Gazing at Rewards: Eye Movements as a Lens into Human and AI Decision-Making in Hybrid Visual Foraging." Proceedings of the Computer Vision and Pattern Recognition Conference. 2025.

**Reviewer Concerns:**

The reviewers raised several concerns.

First, Reviewer 1qs9 noted that some claims in the Introduction were overstated, and multiple reviewers pointed out presentation issues, including figure mismatches, inconsistent naming of ablation settings, and suboptimal section organization.

Second, Reviewers rNCf and wFph highlighted that key aspects of the methodology were insufficiently explained, particularly the definitions and roles of the reward function, the retinal/LGN modeling, and related biological components.
Mutliple reviewers requested a clearer and more systematic explanation of the model architecture and training pipeline, including how gaze supervision is incorporated, how scanpaths are represented, and how attention trajectories are generated and evaluated.

Finally, reviewers expressed concerns regarding experimental coverage and baselines. In particular, the use of CIFAR-10 and ImageNet-10 was considered insufficiently convincing, the interpretability claims were not fully supported by the current experiments, and the DeepGazeIII baseline was deemed inadequate for fair comparison.

The first and second concerns were addressed successfully; however, the third point remains outstanding to AC.

**Reviewer Scores:**

Reviewer 1qs9 may maintain their score, as their concerns have largely been addressed, but they do not appear strongly enthusiastic about the work and are therefore likely to give a weak acceptance rating. The remaining three reviewers are expected to keep their scores unchanged, as their concerns were only partially addressed and the outstanding issues—particularly those related to the third concern above—were raised by all of them and remain unresolved.

---

### Decision · Program_Chairs · 2026-01-26

Reject